



# Satellite Passive Microwave Sea-Ice Concentration Data Set Inter-comparison for Arctic Summer Conditions

Stefan Kern[1], Thomas Lavergne[2], Dirk Notz[3], Leif Toudal Pedersen[4], and Rasmus Tonboe[5]

[1]Integrated Climate Data Center (ICDC), Center for Earth System Research and Sustainability (CEN), University of Hamburg, Hamburg, Germany
[2]Research and Development Department, Norwegian Meteorological Institute, Oslo, Norway
[3]Institute for Marine Research, University of Hamburg and Max-Planck Institute for Meteorology, Hamburg, Germany
[4]Danish Technical University, Lyngby, Denmark
[5]Danish Meteorological Institute, Copenhagen, Denmark

*Correspondence to*: Stefan Kern (stefan.kern@uni-hamburg.de)

**Abstract.** We report on results of a systematic inter-comparison of 10 global sea-ice concentration (SIC) data products at 12.5 to 50.0 km grid resolution from satellite passive microwave (PMW) observations for the Arctic during summer. The products are compared against SIC and net ice-surface fraction (ISF) – SIC minus the per-grid cell melt-pond fraction (MPF) on sea ice – as derived from MODerate resolution Imaging Spectroradiometer (MODIS) satellite observations and observed from ice-going vessels. Like in Kern et al. (2019), we group the 10 products based on the concept of the SIC retrieval used. Group I consists of products of the EUMETSAT OSI SAF and ESA CCI algorithms. Group II consists of products derived with the Comiso bootstrap algorithm and the NOAA NSIDC SIC climate data record (CDR). Group III consists of ARTIST Sea Ice (ASI) and NASA Team (NT) algorithm products and group IV consists of products of the enhanced NASA Team algorithm (NT2). We find wide-spread positive and negative differences between PMW and MODIS SIC with magnitudes frequently reaching up to 20-25 % for groups I and III and up to 30-35 % for groups II and IV. On a pan-Arctic scale these differences may cancel out: Arctic average SIC from Group I products agrees with MODIS within 2-5 % accuracy during the entire melt period from May through September. Group II and IV products *over-estimate* MODIS Arctic average SIC by 5-10 %. Out of group III, ASI is similar to group I products while NT SIC *under-estimates* MODIS Arctic average SIC by 5-10 %. These differences, when translated into the impact computing Arctic sea-ice area (SIA), match well with the differences in SIA between the four groups reported for the summer months by Kern et al. (2019). MODIS ISF is systematically over-estimated by all products; NT provides the smallest (up to 25 %) over-estimations, group II and IV products the largest (up to 45 %) over-estimations. The spatial distribution of the observed over-estimation of MODIS ISF agrees reasonably well with the spatial distribution of the MODIS MPF and we find a robust linear relationship between PMW SIC and MODIS ISF for group I and III products during peak melt, i.e. July and August. We discuss different cases taking into account the expected influence of ice-surface properties other than melt ponds, i.e. wet snow and coarse grained snow / refrozen surface, on PMW observations used in the SIC retrieval algorithms. Based on this discussion we identify the mismatch between the actually observed surface properties and those represented by the ice tie points as the most likely reason for i) the observed differences between PMW SIC and MODIS ISF and for ii) the often surprisingly small difference between PMW and MODIS SIC in areas of high melt-pond fraction. We conclude that all 10 SIC products are highly inaccurate during summer melt. We hypothesize that the unknown amount of melt-pond signatures likely included in the ice tie points plays an important role – particularly for groups I and II – and suggest to conduct further research in this field.

## 1 Introduction

A considerable number of different algorithms to compute the sea-ice concentration from satellite passive microwave (PMW) brightness temperature (TB) measurements has been developed during the past decades. All exploit the fact that under typical viewing angles (50-55 degrees) the difference in microwave TB between (liquid) open water and sea ice is sufficiently large to estimate sea-ice concentration.





In the Polar Regions, freezing conditions prevail during winter. During summer, melting conditions prevail or at least coexist with freezing conditions. The changes in snow and sea-ice properties in response to the melting conditions complicate

the retrieval of the sea-ice concentration from microwave TB measurements. This applies in particular to the Arctic. First signs of melt are an increase in snow wetness and melt-refreeze cycles, triggered by diurnal warming and nocturnal cooling, leading to an increase in snow grain size and snow density. Wet snow is a good absorber of microwave radiation and has an emissivity close to 1. Therefore, microwave TBs measured over wet snow are often very close to the physical temperature of the melting snow, i.e. 0 °C. As a consequence, wet snow masks the radiometric difference between different ice types, e.g. first-year and

multiyear ice. Above a certain wetness and snow thickness (a few centimetres), the influence of wet snow on microwave TBs in the frequency range used here (see Table 1) can be regarded as being independent of frequency and polarization. The influence of coarse-grained snow is more complex. During diurnal melting, it behaves like wet snow. During nocturnal cooling, the liquid water refreezes and considerably less microwave radiation is absorbed. This allows for volume scattering from within the snow, which – in contrast to the absorption of microwave radiation by wet (coarse-grained) snow is both frequency

and polarization dependent. More details about the influence of these parameters on microwave TBs relevant for retrieval of sea-ice concentration during summer are given, e.g., in Kern et al. (2016).

Continued melting results in increasing snow wetness until it becomes saturated with melt water – at which stage melt ponds start to form. The fraction of the ice surface covered by melt ponds formed from melting snow and sea ice typically varies between 10 and 40 % but it can exceed 50 %, e.g., early in the melt season or on land-fast sea ice (e.g. Webster et al.,

2015; Divine et al., 2015; Landy et al., 2014). The fraction of liquid water due to melt ponds on the sea ice poses a particular challenge for the sea-ice concentration retrieval using microwave TB measurements because the penetration depth of microwaves in water at the frequencies listed in Table 1 is of the order of 1 mm (Ulaby et al., 1986). Thus, a water layer with a depth of only a few millimetres is sufficiently opaque to block the thermal microwave emission of the sea ice underneath completely. In addition, the emissivity of fresh water in the ponds and the emissivity of saline water in the leads are the same

at most of the microwave frequencies that we use here (freq. > 10GHz). Therefore, during summer liquid water in form of melt ponds on the sea ice is indistinguishable from liquid water in the cracks and leads between the ice floes in the microwave frequency range used here (e.g. Gogineni et al., 1992; Grenfell and Lohanick, 1985). This has direct consequences for the sea-ice concentration retrieval using satellite TB measurements.

Several studies have revealed various degrees of under-estimation of the sea-ice concentration during summer

conditions in the Arctic (e.g. Ivanova et al., 2015; Rösel et al., 2012b; Markus and Dokken, 2002; Comiso and Kwok, 1996; Steffen and Schweiger, 1991; Cavalieri et al., 1990). A natural explanation of this observed under-estimation would therefore be that those satellite products are rather a good measure of the "net ice surface fraction", that is 1 minus the area fraction of all surface water in the satellite field-of-view. We illustrate the typical summer sea-ice concentration retrieval by a simple example. Consider two grid cells A and B observed during the summer melt season. Grid cell A has 100 % sea-ice cover with

40 % melt-pond fraction. Grid cell B has 75 % sea-ice cover with 15 % melt ponds. Based on physical principles a sea-ice concentration retrieval algorithm should provide a value of 60 % in both cases, i.e. the so-called "net ice surface fraction". There is evidence from literature (e.g., Comiso and Kwok, 1996; Kern et al., 2016) that this is however not the case. It is rather very likely that an algorithm would provide a value of, for instance, 85% for both grid cells A and B, because melt-induced changes in the surface emissivitiy of the visible part of the sea ice are often insufficiently taken into account, yielding to an

over-estimation of the actual net ice surface fraction. Providing a value of 85 % this algorithm would under-estimate the actual sea-ice concentration in grid cell A by 15 % while it would over-estimate it in grid cell B by 10 %. If we interpret the provided value as a net ice surface fraction, it is an over-estimation by 25 % in both cases. In other words, for this quite typical example the retrieved value is highly inaccurate and biased compared to either the actual sea-ice concentration or the net ice surface fraction. The magnitude of this bias is largely unknown and it appears not to be reflected by an appropriate increase in retrieval

uncertainty estimates which – when at all provided – are a measure of the precision, i.e. the interval within which the reported



sea-ice concentration estimate typically varies, and not of the bias. In Fig. 1 we show the seasonal cycle of the sea-ice concentration algorithm standard error – aka the precision – of the OSI-450, SICCI-25km and SICCI-50km products (Lavergne et al., 2019) for illustration. To summarize: 1) we do not know what the sea-ice concentration algorithms actually measure during summer (actual sea-ice concentration or net sea-ice surface fraction), and whichever they measure the accuracy is poor

compared to the winter conditions.

The unknown accuracy makes it difficult if not impossible to use summer satellite PMW sea-ice concentration maps and sea-ice area (SIA) for the evaluation of numerical models (e.g., Notz, 2014; Burgard et al., 2019), or to assimilate such data into numerical models for a quantitative improvement of, e.g., sea-ice forecast for shipping (e.g., Melia et al., 2017). As a consequence, studies about the long-term development of the Arctic sea-ice cover prefer to use sea-ice extent (SIE) over

SIA. The SIE is the sum of the area of all grid cells with a sea-ice area fraction above a certain threshold – usually 15 %. The SIA is the same sum but weighted with the actual sea-ice area fraction. Consequently, biases in the sea-ice concentration certainly have a small influence on the summer SIE while the impact on SIA can be quite large (see for example Kern et al. 2019 Fig. 6 and Figs. G2 and G3). However, it has been found that the SIE and its trend provide a limited metric for the performance of numerical models (e.g., Notz, 2014), and that prediction of the minima in September Arctic SIA and SIE would

benefit from giving more weight to SIA (e.g. Petty et al., 2018). One way to overcome the SIA biases in summer could be to focus on its trends (as opposed to its absolute value) (e.g. Comiso et al., 2017; Ivanova et al., 2014) but this cannot be the solution since there is no guarantee that these biases are stable along the whole time series.

With this study, we aim to give more information about the accuracy of current satellite PMW sea-ice concentration products during summer. We present a systematic inter-comparison of 10 satellite PMW sea-ice concentration products (see

Sect. 2, and Kern et al., 2019) with independent estimates of the summertime Arctic sea-ice concentration, net ice surface fraction, and melt-pond fraction derived from observations of the MODerate resolution Imaging Spectroradiometer (MODIS) aboard the Earth Observation Satellite (EOS) TERRA (Rösel et al., 2011, 2012a). We show the pan-Arctic sea-ice concentration biases with respect to MODIS sea-ice concentration and ice surface fraction for the 10 products for the period 2003 through 2011, illustrate the spatiotemporal variability of these biases, and quantify the biases as a function of melt season

progress and melt-pond fraction. We describe the data and inter-comparison methods used in Sect. 2. In Sect. 3 we give an overview about the pan-Arctic results of our inter-comparison. Section 4 focuses on more detailed comparisons to MODIS sea-ice concentration and ice surface fraction, and ship-based observations of sea-ice concentration and melt-pond fraction, and illustrates the potential of a bias correction as well as the impact on the computation of the sea-ice area. Our paper closes with a discussion and concluding remarks in Sect. 5.

**2    Data & Methodologies**

**2.1    Sea-ice concentration data sets**

Like in Kern et al. (2019), we consider 10 different sea-ice concentration products which we very briefly summarize in Table 2.1.1. More information about these products is given in Kern et al. (2019, Appendix 7.1 -7.6). There are many more algorithms and products available than we are using here, see e.g. Ivanova et al. (2015). The main criteria for our choice of

algorithms and products are 1) length of the product time series, 2) grid resolution, 3) accessibility and sustained extension, and 4) overlap with the melt-pond fraction evaluation data set. Due to these criteria we have not selected products with < 10 years coverage or with a grid resolution < 12.5 km.

In the following few paragraphs we provide some general remarks to the satellite PMW data products used. We refer to Lavergne et al. (2019) and Kern et al. (2019) for further information.

The difference in microwave TBs observed over open water (low) and land (high) combined with the size of the field-of-view of several kilometres to a few tens of kilometres can cause spurious sea-ice concentrations to appear along coasts (e.g.



Lavergne et al., 2019). In this paper, we do neither further correct potential differences between the 10 products caused by this effect nor do we pay particular attention to this effect.

Atmospheric moisture and wind-induced roughening of the ocean surface can cause spurious sea-ice concentrations in areas that are actually ice free. To mitigate this noise, different kind of weather filters are applied in the 10 products used. These and their effects on SIA and SIE estimated from the sea-ice concentration data are discussed in Kern et al. (2019). The focus of this paper is on the performance of the 10 products during summer conditions over consolidated ice, where the weather filters have no effects. Therefore, we do not further discuss weather filters in this paper.

In near-100 % and near-0 % sea-ice concentration conditions, most retrieval algorithms will naturally retrieve a bell-shaped distribution of sea-ice concentration values, returning values both below and above 100 % or 0 % sea-ice concentration (e.g. Ivanova et al., 2015). However, the above-100 % values are almost never accessible to the user, and thus generally not accessible for validation. While the EUMETSAT-OSISAF – ESA-CCI products (group I, see Table 1) allow using the naturally retrieved sea-ice concentration on either side of 100 % the others do not. In those other products any sea-ice concentration values retrieved as being > 100% are set to 100 % and lost; these values are not accessible to the user. The availability of these

"off-range" estimates in the four group I products was used in Kern et al. (2019) to demonstrate how the "off-range" distribution can effectively be reconstructed a-posteriori for most of the other products from their truncated sea-ice concentration distributions. Kern et al. (2019) illustrated that products with over-estimated sea-ice concentration (modal value of the non-truncated distribution larger than 100 %) would obtain better validation statistics (smaller bias and RMSE) than products with no over-estimation (modal value of the non-truncated distribution exactly at 100 %). The larger the over-

estimation, the better the statistics would be. In Sect. 5.1.1 of this paper, we will get back to this issue and its relevance for our comparison with the MODIS data set.

## 2.2    The MODIS data set

We use the MODerate resolution Imaging Spectroradiometer (MODIS) Arctic melt-pond fraction data set developed by Rösel et al. (2011; 2012a): Rösel et al. (2015), https://doi.org/10.1594/WDCC/MODIS__Arctic__MPF_V02, last accessed

October 12, 2016. This data set is provided for the Arctic Ocean north of 60°N with 8-daily temporal resolution on the NSIDC polar-stereographic grid with 12.5 km x 12.5 km grid resolution at 70 degrees north. It extends from day-of-year (DOY) 129, i.e. May 9 (May 8 for leap years), to DOY 256, i.e. September 13 (September 12 for leap years) and hence covers pre-melt, melt advance, peak melt and end-of-melt conditions.

The melt-pond fraction retrieval is based on the calibrated and atmospherically corrected reflectance values measured

by MODIS channels 1, 3 and 4 available in the MOD9A1 8-day product. For this product, reflectance values measured during eight consecutive days were re-projected from the original MODIS tiles into the NSIDC polar stereographic grid with 500 m x 500 m grid resolution, composited over the 8-day period, and combined with the cloud- and land masks provided with the MOD09 product. For the retrieval, it is assumed that each 500 m grid cell is solely covered by fractions of three surface types: open water in leads and openings between ice floes, melt ponds, and sea ice and snow. The sum of these fractions is assumed

to equal 1. Via a spectral un-mixing approach and an artificial neural network the measured reflectance values are converted into the fractions of these three surface types per grid cell, followed by the interpolation onto the 12.5 km grid used for the final product. The product has undergone various levels of evaluation, e.g. comparison with ship-based observations and comparison with air-borne optical imagery. For more details about the method and the evaluation with air-borne data, we refer to Rösel et al. (2012a).

The product contains the melt-pond fraction (MPF), the open water fraction (OWF), the standard deviation of the MPF values at 500 m grid resolution, and the number of valid 500 m MPF estimates. This latter number is a measure of the number of clear-sky 500 m grid cells. In addition the product contains so-called "clear-sky" versions of the 12.5 km gridded MPF and OWF data computed only for those 12.5 km grid cells where more than 90 % of the input 500 m grid cells are denoted





clear-sky. We note that the MPF is a measure of the melt-pond fraction per grid cell. No MPF values are provided for 12.5 km
grid cells with an OWF larger than 85 %.

Rösel et al. (2012a) reported root-mean-squared errors (RMSE) between ~4 % and 11 % compared to air-borne data.
Kern et al. (2016) compared daily estimates of the MPF for June to August 2009 with ship-based observations of the MPF and
reported RMSE values between ~6 % and ~15 %. Istomina et al. (2015) and Marks (2015) confirmed the validity of the MODIS
MPF data set with different independent observations of melt ponds. Experience working with this data led us to conclude that
the MPF estimates are accurate to within a few percent. However, besides unaccounted cloud influence there is another
limitation that needs to be kept in mind when using this data set. The used approach is based on three channels, which limits
the maximum number of surface types to be discriminated to three. Ponds on first-year ice, however, have different spectral
characteristics than ponds on multiyear ice: while the latter appear and remain bluish and relatively bright, the former become
darker with advancing melt season until they eventually melt through the ice. As a consequence, towards the end of the melt
season melt ponds on first-year ice might be assigned to the class open water. During the same time of the melting season,
melt ponds might be covered by a slush or thin ice layer. Depending on the properties of this layer, the melt pond is either still
assigned to the class melt pond, or it is assigned to the class ice. In addition, new ice forming between the ice floes in the high
Arctic towards the end of the melt season could be classified as melt ponds (see also Rösel et al., 2012a). Because of these
ambiguities in the retrieval of the melt-pond fraction it is likely that the accuracy of the parameters derived is poorer towards
the end of the melt season, i.e. September.

For this paper, we used the clear-sky versions of MPF and OWF. In addition, we exclude all those MODIS data set
grid cells where the ratio between the 12.5 km gridded MPF value and the standard deviation of the 500 m MPF values is
larger than 1. While this step filters out grid cells with an actual true large MPF variability, at the same time it reduces the
influence of cloud cover artefacts. A similar filtering effect could have been achieved by increasing the percentage of 500 m
grid cells required to consider a 12.5 km grid cell value clear-sky from 90 % to, for instance, 95 %. However, in that case the
number of valid MODIS product data would have reduced drastically.

We are interested in the fraction of ice detectable with PMW sensors. We call this the net ice surface fraction (ISF).
ISF is related to OWF and MPF as follows: ISF + MPF + OWF = 1. We thus derive two parameters from the MODIS data set:
the MODIS sea-ice concentration, MODIS SIC, which is 1 – OWF, and the MODIS ice surface fraction, MODIS ISF, which
is 1 – OWF – MPF.

Sea-ice concentrations of the 10 products (Sect. 2.1) are co-located with the MODIS parameters via finding the grid
cell pairs with the minimum difference (in kilometres) between the grid-cell centres. For this step the coordinates of both data
sets, i.e. the PMW products and the MODIS products, are converted into Cartesian coordinates allowing to compute the
minimum distance via simple geometry. We do not interpolate any of the data sets. We do not perform any averaging in case
that multiple (small) grid cells of one product fall into one (large) grid cell of the other product. All comparisons are carried
out at the native grid resolution. When compared to the 25 km products, this results in a lower number of co-located grid cells
for SICCI-50km and a higher number for SICCI-12km and ASI-SSMI. Finally, the collocated PMW SIC data are averaged in
time over the same eight days used in the respective 8-daily MODIS product, i.e. for the MODIS product of DOY=129 we
average over data from DOY 129 through 136. If valid sea-ice concentrations of fewer than three days within this 8-day period
are available, this grid cell is discarded from further analysis.

In Fig. 2 we illustrate the melt-pond development in the Arctic Ocean as relevant for this paper. The maps show melt-
pond distributions of a DOY representative of the four periods considered in this paper: pre-melt, melt advance, peak melt and
end-of-melt in the maps of Fig. 2a-d, respectively. For these maps, we selected years where the data coverage is particularly
good, i.e. with only few grid cells discarded as potentially cloud contaminated. Below each map we show histograms of the
melt-pond fraction of the respective DOY of years 2003 to 2011, to illustrate the inter-annual variability.



### 2.3     Ship-based visual sea-ice cover observations

For our inter-comparison of the 10 products, we also used ship-based manual visual observations of the summer-time sea-ice conditions collected under the IceWatch/ASSIST (Arctic Ship-based Sea-Ice Standardization, see http://icewatch.gina.alaska.edu or https://icewatch.met.no/). Such observations, carried out hourly from the ships' bridge while the ship navigates through the sea ice, provide information about, e.g., total and partial sea-ice concentrations, status of the ice surface and melt-pond concentration. We refer to Worby et al. (2008) and Hutchings and Orlich (2019) for more details. Ship-based observations of the ice conditions have been used already in the past to evaluate PMW SIC products in the otherwise data sparse Arctic region (e.g., Alekseeva et al., 2019; Kern et al., 2019; Wang et al., 2018; Xie et al., 2013; Spreen et al., 2008). We use observations provided in a standardized format from https://doi.org/10.26050/WDCC/ESACCIPSMVSBSIO, last access date: 28 October, 2019. Standardization means that the resulting ascii format data file containing all observations uses similar formats for all variables and missing data. The data are also manually quality checked for outliers. We use all observations during months May through September, for the period June 2002 through December 2011. Figure 3 illustrates the location of the ship tracks from which we used such observations, colour coded with respect to the year of observation. Note that while observations of the sea-ice concentration were made throughout all these tracks, observations of melt ponds are sparser and only available for about one third of the locations shown in Fig. 3.

We co-locate the sea-ice concentrations of the 10 products with the selected ship-based observations by computing the minimum distance between geographic location of the ship-based observation and the grid cell centre of the respective sea-ice concentration product at its native resolution. For this step, we convert the geographic coordinates of all data sets into Cartesian coordinates taking into account the different projections of the sea-ice concentration products. After co-location, we compute daily along-track averages of the ship- and satellite-based sea-ice concentrations following the approach of Beitsch et al. (2015). We discard data pairs with less than three observations per day.

### 3     Pan-Arctic summer-time sea-ice conditions

We begin our inter-comparison with an illustration of the sea-ice conditions in the Arctic during summer as seen by the satellite products. For this step, we compute an ensemble multi-annual (2002-2011) median of the monthly mean sea-ice concentration from the 10 sea-ice concentration products and subtract it from the respective sea-ice concentration of the individual product. This computation is carried out at 50 km grid resolution using a common land mask (see Kern et al., 2019). In Fig. 4 we illustrate the full suite of differences between individual products and the ensemble median as an example for the month of July (Fig. 4a-j) along with a map of the ensemble median sea-ice concentration (Fig. 4k). Within groups I, II and IV the products exhibit a similar distribution of sea-ice concentration differences. These are mostly negative for group I (mean sea-ice concentration smaller than ensemble median) but mostly positive for groups II and IV (mean sea-ice concentration larger than ensemble median). By far the largest negative difference we find for NT1-SSMI from group III (Fig. 4i). We refer to Kern et al. (2019) for an explanation of large negative differences along some coastlines shown for SICCI-50km (Fig. 4c).

Differences between the individual products' multi-annual pan-Arctic monthly mean sea-ice concentration and the ensemble median increase from winter (Table 2, top row, Jan/Feb) to summer (Table 2, bottom row, July/Aug). Group I products show less sea ice than the ensemble median; group II and group IV products show more sea ice than the ensemble median; this applies to winter and summer. The absolute sea-ice concentration differences between the individual products and the ensemble median increase from winter to summer – except for ASI-SSMI.

These findings document, together with the results presented and discussed in Kern et al. (2019, Fig. 11 and Appendix G), that the 10 PMW SIC distributions differ considerably in summer. These findings agree with results from previous inter-comparisons of PMW SIC products (e.g., Comiso et al., 2017; Ivanova et al., 2014, 2015; Spreen et al., 2008; Meier, 2005).



## 4  Results

### 4.1  Inter-comparison against MODIS sea-ice concentration (SIC)

Figure 5 exemplifies how the difference: PMW SIC minus MODIS SIC changes as a function of the stage of melt for the four groups of products. For this illustration, we select the same 8-day periods as used in the maps shown in Fig. 2. The four

rows represent stages of melt: pre-melt (DOY 129, May 9-16), melt-advance (DOY 169, June 18-25), peak melt (DOY 201, July 20-27) and end of melt (DOY 241, Aug. 29-Sep. 5). The four columns represent the four groups of products by showing results for OSI-450 (group I), CBT-SSMI (group II), NT1-SSMI (group III) and NT2-AMSR-E (group IV). These examples are taken from different years, chosen because of a relatively small number of invalid or missing data. Figure 6 shows two-dimensional (2-D) histograms of PMW SIC (y-axis) versus MODIS SIC (x-axis) corresponding to the SIC maps used for the

differences shown in Fig. 5. Similar histograms but based on SIC data of years 2003 to 2011 are provided in the supplementary Figure S1. We omit the pre-melt examples in Figure 6 (and Figure S1) because they exhibit limited additional information. These are shown for completeness in the supplementary Figure S2. Overall, we find a similar development of SIC differences in the maps (Fig. 5) as well as in the 2-D histograms (Figs. 6, S1 and S2) across the four stages of melt for groups I and III on the one hand, and for groups II and IV on the other hand.


### 4.1.1  *Pre-melt*

For the pre-melt example (Fig. 5a-d), group II and IV products slightly and more or less uniformly over-estimate MODIS SIC (Fig. 5b, d). Group I and III products exhibit areas of over- and under-estimation of MODIS SIC (Fig. 5a, c). The latter occur north of the Laptev Sea and the Fram Strait and are more pronounced for group III for which negative differences may

have values ≥ 15 % in magnitude. Given the estimated accuracy of the MODIS SIC, which is about 5 % (see Kern et al., 2016), we can state that for pre-melt PMW SIC and MODIS SIC mostly agree within their uncertainties – except for the mentioned areas with larger under-estimations of MODIS SIC (groups I and III).

### 4.1.2  *Melt advance*

For the melt advance example (Fig. 5e-h), all four groups over-estimate MODIS SIC by 5-10 % south of the pole facing Greenland and the Greenland, Barents and Kara Sea. MODIS SIC is under-estimated for much of the central Arctic Ocean, the Beaufort Sea and towards the East-Siberian Sea by groups I and III. MODIS SIC is over-estimated by groups II and IV for much of the Beaufort and Chukchi Sea. Absolute differences remain mostly < 15 %. Groups I, III and IV appear to have strong gradients in the differences (see e.g. Fig. 5e and f). At this stage, melt has commenced everywhere (Fig. 2 b). The bluish area

with melt-pond fraction (MPF) values < 15 % in Fig. 2b corresponds well to the area exhibiting only small absolute differences for group II (Fig. 5f) or group IV (Fig. 5h). The relatively abrupt change from these low differences in the central Arctic Ocean to positive differences of about 15 % in the Chukchi and Beaufort Sea coincides with a similarly abrupt change in MPF from values < 15 % to values > 25 %. It appears, however, that there is no unique correspondence between SIC differences and the MPF. For instance, near-0 % differences between PMW SIC and MODIS SIC co-exist with MPF values < 10 % and > 30 %

for groups II and IV. Likewise, for groups I and III, the spatial variability of the differences shown in Fig. 5e and g in the central Arctic Ocean is not reflected by the spatial variability in the MPF (Fig. 2b).

The larger tendency for an over-estimation of MODIS SIC for groups II and IV shown in Fig. 5, compared to groups I and III, is also illustrated in the respective 2-D histograms (Fig. 6, left column). For groups II and IV (Fig. 6d and j), we find the highest counts above the identity line (note the logarithmic scale of the count), while for groups I and III (Fig. 6a and g)

we find a more symmetric distribution around the identity line. Over the period 2003-2011 (Fig. S1, left column), all four groups have the majority of SIC value pairs concentrated at > 90 % SIC. However, while groups I and III (Fig. S1a and g)



have a substantial fraction of value pairs below the identity line, concomitant with a linear regression line slope < 1 and a negative intercept, groups II and IV (Fig. S1d and j) have a larger fraction of value pairs above the identity line, a linear regression line slope > 1, and a positive intercept.


### 4.1.3 Peak melt

For the peak melt example (Fig. 5i-l), groups II and IV over-estimate MODIS SIC almost everywhere by up to 20%. This over-estimation is confirmed well by the respective 2-D histograms (Fig. 6e and k). Regions with only 5 to 10 % over-estimation of MODIS SIC by these groups correspond to a range of different MPF value: up to 15 % in the central Arctic Ocean, ≥ 30 % in the southern Beaufort Sea and 40 % in the Canadian Arctic Archipelago (see Fig. 2c). Regions with the highest over-estimation of MODIS SIC exhibit MPFs of 25-30 % in Fig. 2c. Groups I and III (Fig. 5i, k) have similar spatial patterns in the SIC differences which differ considerably from those of groups II and IV. Group I mostly over-estimates MODIS SIC, for group III MODIS SIC under-estimation appears to dominate.

The magnitude of the positive (negative) differences is larger for group I (group III) as also illustrated in the respective 2-D histograms where group III has a larger fraction of values below the identity line and a less steep slope than group I (Fig. 6b and h). Areas with MODIS SIC under-estimation correspond relatively well to areas with MPF > 25 % and areas with MODIS SIC over-estimation mostly exhibit MPF < 20 % (Fig. 2c). Of all groups, group I has the highest linear correlation (0.84) and the smallest root-mean-squared difference (RMSD) of 7.8 % between PMW SIC and MODIS SIC (see Fig. 6, middle column). Over the period 2003-2011 (Fig. S1, middle column), groups I and III provide a quite symmetric distribution with linear correlations of 0.86 and 0.87, respectively. For group IV (Fig. S1k) and especially group II (Fig. S1e) over-estimation of MODIS SIC dominates – in agreement with Fig. 5j and l. Even though the linear correlation of 0.85 for group II is as high as those of groups I and III the distribution of values (Fig. S1e) suggests two separate linear regressions.

### 4.1.4 End-of-melt

For the end-of-melt example (Fig. 5m-p), spatial patterns of MODIS SIC over-estimation by groups II and IV are very similar (Fig. 5n, p) as are the respective 2-D histograms (Fig. 6f, l); almost all values are above the identity line, the majority of PMW SIC is > 90 % and MODIS SIC ranges between 70 % and 100 %. MODIS SIC over-estimation is largest where the melt-pond fraction is largest and vice versa (compare Fig. 5n, p with Fig. 2d) – except in the southern Beaufort Sea: MPFs > 35 % but near-0 % SIC differences. Group III under-estimates MODIS SIC for most of the sea-ice cover by an amount of up to 25 % (Fig. 5o). Group I overall exhibits the smallest differences to MODIS SIC (Fig. 5m). This is confirmed by Fig. 6c showing the most symmetric SIC distribution around the identity line and the smallest RMSD of 7.8 % of all groups. Over the period 2003-2011 (Fig. S1, right column), groups I and III have a quite symmetric distribution similar to peak melt. In contrast, for groups II and IV, the majority of values is concentrated at PMW SIC > 95 % while MODIS SIC is 80 to 100 % (Fig. S1f and l).

325

### 4.1.5 Summary of the comparison to MODIS SIC

We summarize the average values of the statistical parameters: regression line slope and intercept (or offset), linear correlation and RMSD in Table 3. The averages are computed separately for the four stages of melt as the arithmetic mean over all parameter values of the respective group's products and 8-day periods within years 2003 to 2011. For example, for the average RMSD of group I for pre-melt, we average over four (products in group I) times three (three 8-day periods within pre-melt: DOY 129, 137 and 145) times nine (years) values. We do not further interpret the values given for pre-melt and refer the reader to the supplementary Figures S2 and S4. Table 3 shows an increase in correlation, RMSD and slope from melt advance to peak melt for all four groups; towards end-of-melt, correlations decrease and RMSD values increase. Overall,



highest correlations between PMW SIC and MODIS SIC are obtained for groups I and III: 0.75 as a mean over melt advance
to end-of-melt. If we take the RMSD as a measure of how accurate PMW SIC match MODIS SIC, group I products are the
most accurate ones with a mean RMSD from melt-advance to end-of-melt of 8.3 %. Linear regression line slopes are
particularly close to 1 for group II during peak melt and end-of-melt, but the distribution of values in the respective 2-D
histograms (Fig. S1) does not necessarily support a linear functional relationship.

Figure 7 summarizes our results about the pan-Arctic (Arctic Ocean and Canadian Arctic Archipelago) multi-annual
mean melt-season development of PMW SIC of the 10 products in comparison to MODIS SIC and MPF. Temporal sampling
is eight days. The mean MODIS SIC (blue triangles) first stays constant at about 95 % until about DOY 160, i.e. the 2nd week
of June. Subsequently it begins to decrease reaching a minimum of 81 % at around DOY 230, i.e. the 3rd week of August. The
mean MODIS MPF is < 3 % during the first half of May, gradually increasing to a mean MPF of 20-25 % between DOY 180
and DOY 235, i.e. between end of June and the 3rd/4th week of August (this corresponds to peak melt). We find the smallest
difference between PMW SIC and MODIS SIC for group I (Fig. 7a-d), orange symbols). These near-0 % differences result
from positive and negative biases with magnitudes up to 15 % cancelling out (see e.g. Fig. 5e, g, h and k, Fig. 6b, c, and Fig.
S1b, c). Group II products (Fig. 7e-g) exhibit near-0 % differences until the end of June but show up to 10 % *more* sea ice than
MODIS afterwards – in line with widespread positive biases (e.g. Fig. 5j, l) and asymmetric distributions in the respective 2-
D histograms (Fig. 6e, f, k and l, and Fig. S1e, f, k and l). NT1-SSMI (group III, Fig. 7j) first exhibits differences close to zero
but shows *less* sea ice than MODIS during peak melt when the pan-Arctic difference is ~ -5 % increasing in magnitude towards
end-of-melt; this is in line with the observed widespread negative biases (Fig. 5g, k, o) and an under-estimation of MODIS
SIC which increases as SIC decreases (Fig. 6h, i, and Fig. S1h, i).

Of the four groups of products investigated, we get three different kinds of agreement between PMW SIC and MODIS
SIC. Most importantly, instead of an under-estimation of the actual sea-ice concentration by PMW SIC (e.g. Rösel et al, 2012b:
Comiso and Kwok, 1996; Steffen and Schweiger, 1991; Cavalieri et al., 1990) our results suggest that an over-estimation of
the actual sea-ice concentration is often more common than an under-estimation. With that this study agrees with the findings
in Kern et al. (2016) but we note that the latter study is based i) on data of one summer season only; ii) on data of a sub-region
of the Arctic Ocean only; iii) on PMW SIC values computed on our own from passive microwave TB measurements using
winter ice-tie points and allowing SIC values > 100 %. Our results in this paper reveal substantial differences between group
II products, e.g. CBT-SSMI, and group III products, e.g. NT1-SSMI, both in magnitude as well as spatial distribution of
differences between PMW SIC and MODIS SIC.

### 4.2   Inter-comparison against MODIS net ice-surface fraction (ISF)

Melt ponds have the largest impact on PMW SIC because of the inability – at the microwave frequencies used – to
discriminate between open water in the form of melt ponds on the ice floes and open water in the form of leads and openings
between the ice floes. Like detailed in the introduction, based on physical principles a sea-ice concentration retrieval algorithm
should provide a value of 60 % for a case A: 100 % sea-ice concentration with 40% melt ponds, and a case B: 75% sea-ice
concentration with 15 % melt ponds; in other words the algorithm should provide the net ice-surface fraction. Therefore, a
logical next step to better understand the causes of the SIC differences reported in Sect. 4.1 is to investigate how PMW SIC
compares to ISF, as derived, e.g., from MODIS (Sect. 2.2). The inter-comparison between PMW SIC and MODIS ISF is
carried out similarly to the inter-comparison to MODIS SIC (Sect. 4.1). We organize the results exactly in the same structure
as in Sect. 4.1. We present a set of maps of the differences PMW SIC minus MODIS ISF for selected 8-day periods (see Fig.
2) in Fig. 8 (compare Fig. 5), complemented by the respective 2-D histograms shown in Fig. 9 (compare Fig. 6) and extended
to the entire period 2003-2011 in Fig. S3 (compare Fig. S1).



### 4.2.1    Pre-melt

For the pre-melt example (Fig. 8a-d), practically no melt ponds exist in the Arctic Ocean (see Fig. 2a); MODIS ISF equals MODIS SIC. Differences PMW SIC minus MODIS ISF shown in Fig. 8a-d are almost identical to the maps shown in Fig. 5a-d. The differences in the 2-D histograms (Supplementary Figs. S2 and S4, left column) are small. However, considering the entire period 2003-2011 (Supplementary Figs. S2 and S4, right column) we find a tail of near-100 % PMW SIC values which spreads over a range of MODIS ISF between 60-70 % and 100 %; this tail is more pronounced for groups II and IV. The values in this tail are from locations where MODIS ISF < MODIS SIC and PMW SIC over-estimates MODIS ISF, e.g. in the Laptev Sea, the East Siberian Sea and for groups II and IV in Lancaster Sound and Kotzebue Sound also (Fig. 8a-d). At these locations MPF is ~ 10 % (Fig. 2a).

### 4.2.2    Melt advance

For the melt advance example (Fig. 8e-h), we find widespread over-estimation of MODIS ISF by all groups. Highest over-estimations occur in the Chukchi and Beaufort Sea: up to 25 % for groups I and III (Fig. 8e, g), up to 35 % for groups II and IV (Fig. 8f, h). North of this region an area extending across the Arctic Ocean towards Fram Strait has the smallest over-estimation of MODIS ISF: 5-10 % for groups II and IV, 0-10 % for groups I and III. We note patches of under-estimation of MODIS ISF by up to 15 % for group III (Fig. 8g). Overall the spatial pattern of differences PMW SIC minus MODIS ISF matches reasonably well with the respective MPF map (Fig. 2b): The area with differences ≤ 10 % in the central Arctic Ocean coincides with a MPF of ~10 %. The area with high differences in the Chukchi and Beaufort Sea coincide with MPF of 25-35 %. We note, however, that areas with highest over-estimation of MODIS ISF (e.g. groups II and IV) do not necessarily coincide with the highest MPF. We get back to this issue in Sect. 5.1.2.

The respective 2-D histograms (Fig. 9, left column), reveal a similar distribution of the value pairs and magnitude of counts for groups I and III (Fig. 9a, g) on the one hand, and groups II and IV (Fig. 9d, j) on the other hand – like we found in Sect. 4.1 for MODIS SIC. Common to all 2-D histograms for melt advance is a bi-modal distribution of counts with modes approximately centred at value pairs of PMW SIC ~100% / MODIS ISF ~90% and at PMW SIC ~80 % SIC / MODIS ISF ~60 % for groups I and III. For groups II and IV the second mode is located at higher PMW SIC: ~95 % SIC. The locations of the modes agree well with the differences shown in Fig. 8e-h. We note that values of the linear correlation coefficient (RMSD) are higher (lower) for groups I and III compared to groups II and IV. The findings from Fig. 9, left column, appear to be typical for the entire period 2003-2011 as illustrated by the 2-D histograms shown in Fig. S3, left column. These histograms extend the view given by Fig. 9, left column, in all aspects.

### 4.2.3    Peak melt

For the peak melt example (Fig. 8i-l), PMW SIC over-estimates MODIS ISF everywhere. The over-estimation is particularly high for group II (Fig. 8j): 20-25 % in the central Arctic Ocean and up to 45% in the Chukchi and Beaufort Sea. The over-estimation is lowest for group III (Fig. 8k): 20-25 % in most areas and smaller values in the Canadian Arctic Archipelago and towards the ice edge. The spatial distribution of the differences matches relatively well with the observed MPF (Fig. 2c): around 25 % over most of the Arctic Ocean, down to 15 % in the central Arctic and up to 35 % towards the ice edge and the Canadian Arctic Archipelago.

The respective 2-D histograms (Fig. 9, middle column) reveal – like for melt advance – similar distributions of value pairs and counts for groups I and III (Fig. 9b, h) on the one hand, and groups II and IV (Fig. 9e, k) on the other hand. For groups I and III, linear regression lines almost parallel the identity line and high counts distribute relatively homogeneous among the values down to PMW SIC: 65-70 % and MODIS ISF: 40-45 % (Fig. 9b). For groups II and IV, in contrast, high counts concentrate at PMW SIC: 95-100 % and ~95 %, respectively, and range over MODIS ISF: 55-75 % (Fig. 9e) and 60-





75 % (Fig. 9k), respectively. MODIS ISF values of 40-45 % are associated with group II PMW SIC of 80-95 % and group IV PMW SIC of 75-90%, in line with the high differences observed in Fig. 8j and l. The highest linear correlation coefficient and lowest RMSD are obtained for groups I and III. The findings from Fig. 9, middle column, appear to be typical for the entire period 2003-2011 as illustrated by the 2-D histograms shown in Fig. S3, middle column. These histograms extend the view

given by Fig. 9, middle column, in all aspects and suggest a solid linear relationship between PMW SIC and MODIS ISF for groups I and III. We get back to these linear relationships in Sect. 4.4.

### 4.2.4    *End-of-melt*

For the end-of-melt example (Fig. 8m-p), we find reasonable agreement between the distribution of the MPF (Fig. 2d) and the difference PMW SIC minus MODIS ISF for all groups. Areas of MPF < 5 % coincide with differences between -10 %

and 10 %. Groups II and IV mostly over-estimate MODIS ISF (Fig. 8n, p) while group III mostly under-estimates MODIS ISF (Fig. 8o). Areas with high MPF not necessarily coincide with areas of a large difference PMW SIC minus MODIS ISF across the groups. This is illustrated well by the region with MPF ~35 % in the Beaufort Sea (Fig. 2d) for which we find differences of ~15 % (group III, Fig. 8o), ~20 % (group I, Fig. 8m) and ~40 % (groups II and IV, Fig. 8n, p).

The respective 2-D histograms (Fig. 9, right column) reveal very different relationships between PMW SIC and MODIS

ISF for groups I and III on the one hand and groups II and IV on the other hand. For groups I and III, the majority of values and high counts concentrate at 80-100 % for both PMW SIC and MODIS ISF. A tail of substantially smaller counts extends towards lower SIC / ISF values (Fig. 9c, i); this tail is quite close to the identity line for group III. Values and high counts of groups II and IV (Fig. 9f, l) concentrate at high PMW SIC: > 95 % and > 90 %, respectively, but range over MODIS ISF values of 55-100 % and 60-100 %, respectively (compare to Fig. 8n and p). When considering the entire period 2003-2011

(Fig. S3, right column), values scatter much more and areas with high counts are much less confined – at least for groups I and III. Over-estimation of MODIS ISF by PMW SIC appears still to be more pronounced for groups II and IV (see Fig. S3f).

### 4.2.5    *Summary of the comparison to MODIS ISF*

Table 4 shows an increase in mean values of correlation, RMSD and slope, computed as described for Table 3, from melt

advance to peak melt for all four groups. Towards end-of-melt, all parameters decrease. Overall, highest correlations between PMW SIC and MODIS ISF we find for groups I and III: 0.76 as the mean from melt advance to end-of-melt. Group III has the smallest mean RMSD: 19.0 % from melt advance to end-of-melt, and 23.2 % during peak melt; respective values for group I are 19.5 % and 24.5 %. These values can be taken as a measure of MODIS ISF over-estimation by PMW SIC. Slopes of the linear regressions get most close to 1 for groups I and III during peak melt (0.86 and 0.85) suggesting a solid linear relationship

between PMW SIC and MODIS ISF – also in view of the distributions of values and counts in the 2-D histograms.

Agreement between the MPF and the magnitude of the difference PMW SIC minus MODIS ISF differs among the four groups (compare Figs. 2 and 8). It appears that MODIS ISF is over-estimated by group III by an amount smaller than the MPF while for groups II and IV the MODIS ISF over-estimation is often larger than the MPF. This observation is confirmed by Fig. 10 (compare with Fig. 7). For group I (Fig. 10a-d), MPF values (in cyan) agree with the difference PMW SIC minus MODIS

ISF (in orange) within 2% for the entire melt season. Hence, on a pan-Arctic scale, averaged over years 2003 to 2011, group I products' over-estimation of MODIS ISF *equals* the MPF. The over-estimation of MODIS ISF by group II (Fig. 10e-g) and group IV (Fig. 10j) products is *smaller* than the MPF during peak melt and end-of-melt by up to 10 %, while for group III (NT1-SSMI, Fig. 10i) this over-estimation is *larger* than the MPF by up to 10 %. We note that ASI-SSMI, although belonging to group III, performs similar to group IV (compare Fig. 10h and j).




### 4.3    Comparison to ship-based observations

This study so far considered only one source of non-PMW SIC and ISF, namely the dataset of Rösel et al., (2011; 2012a) based on MODIS data. In this section, we report on an alternative source of SIC and ISF information, and assess if comparisons with the new data consolidate our conclusions. Kern et al. (2019) carried out an inter-comparison between the 10

products used here and manual visual ship-based observations of the sea-ice cover. Here we extend this analysis by showing a similar comparison focussing on summer (May through September) of years 2002-2011 (see Sect. 2.3) in Fig. 11. We use ship-based observations of the sea-ice concentration as well as of the net ice-surface fraction. We refer to Kern et al. (2019) for a discussion of the quality and limitations of such observations and note here that our main intention for showing this inter-comparison is to confirm the different results of the inter-comparison between PMW SIC and MODIS SIC (Sect. 4.1) or

MODIS ISF (Sect. 4.2). Figure 11 suggests agreement between PMW SIC and the ship-based SIC observations which is different for groups II and IV (Fig. 11e-g, j) than for groups I (Fig. 11a-d) and III (Fig. 11h, i) – similar to our results in Sects. 4.1 and 4.2. Overall, SIC differences are smaller for groups II and IV than the other two groups while the standard deviation of the differences is smallest for group I products. For group II and IV products, we find that high PMW SIC > 95 % is associated with a considerably larger range of ship-based SIC observations in the scatterplots shown. Nevertheless, the

statistical parameters suggest that CBT-SSMI and NOAA CDR of group II agree well with the ship-based SIC observations while group III products are biased low by between ~ -5 % (ASI-SSMI) and ~ -12 % (NT1-SSMI) and the SICCI/OSISAF products (group I) by ~ -9 %. These differences are in line with the results of the comparison between PMW SIC and MODIS SIC (see e.g. Fig. 7) in the sense that group II products provide the highest PMW SIC, group III the lowest and groups I and IV intermediate SIC values.

The value pairs of PMW SIC and ship-based ISF (red dots) also confirm our observations from Sect. 4.2. For group I and ASI-SSMI of group III (Fig. 11a-d, h), the majority of these value pairs fall into the intervals: 75 % ≤ PMW SIC ≤ 100 % and 50 % ≤ ship-based ISF ≤ 75 %, corresponding to an average over-estimation of the ISF by ~25 %. We note that for NT1-SSMI (Fig. 11i), the value pairs are less clustered and closer to the identity line than for any other product, suggesting a smaller difference between PMW SIC and ship-based ISF. For groups II and IV (Fig. 11e-g, j), most PMW SIC values cluster

close to 100%; accordingly, the majority of these values pairs fall into the intervals: 90 % ≤ PMW SIC ≤ 100 % and 50 % ≤ ship-based ISF ≤ 75 %, suggesting an average over-estimation of the ISF by ~35 %. This order of the magnitude by which the ship-based ISF is over-estimated by the PMW SIC of the different groups is in line with our results from Sect. 4.2 as well.

### 4.4    Bias correction as a potential way forward

The main motivations for this paper are to evaluate the performance of PMW SIC products during summer conditions

and to better understand why PMW SIC products usually do not provide the net ice surface fraction – which they should following physical principles. The results presented so far document that none of the existing PMW SIC products provide a faithful picture of the ISF, nor are accurate measures of the true SIC. The discussion given further below in Sect. 5.1.2 does not only reveal possible explanations of the diversity of evaluation results but demonstrate the complexity involved in a potentially planned improvement of the used algorithms – be it by further development of the algorithm itself, be it via

application of more advanced ice tie-point retrieval approaches. Here we discuss potential ways forward in the short to medium term (using existing PMW products) and in the longer term (preparing and using improved PMW SIC products).

For products of groups I and III, our comparison between PMW SIC and MODIS SIC and between PMW SIC and MODIS ISF suggests linear functional relationships (Sects. 4.1 and 4.2). In the short term, these offer the prospect for users of the existing PMW SIC datasets (especially from groups I and III) to perform bias corrections of the PMW SIC towards either

true SIC (representative of the sea ice area fraction of the geophysical model at hand), or net ISF (representative of all surface water from both ocean and melt ponds). Slope and intercept values prepared in Table 3 and Table 4 allow such bias correction, noting all the limitations of these parameter values that are derived at a pan-Arctic and multi-year scale. For example, we find



values of the linear correlation > 0.85 and slope close to 1 (see Table 4) with respect to MODIS ISF. With such a bias correction one might be able to get closer to the physically more meaningful result of a PMW SIC which equals the net ice surface fraction. The bias correction towards true SIC is somewhat less skilled.


We follow up with this idea and use the linear regression equation obtained for each of the 10 products from the comparison between PMW SIC and MODIS ISF for a bias correction of PMW SIC. We first test how well the bias correction works in comparison to MODIS ISF, i.e. investigate whether the difference between PMW SIC and MODIS ISF is reduced to zero, and subsequently compare the bias-corrected PMW SIC to MODIS SIC. This bias correction is exemplarily carried out

for OSI-450 (group I), CBT-SSMI (group II) and NT1-SSMI (group III) for peak melt (DOY 201, year 2009), shown in Fig. 12 and for melt advance (DOY 169, year 2010), shown in supplementary Fig. S5.

The bias correction works well with respect to MODIS ISF for peak melt. The majority of the differences bias-corrected PMW SIC minus MODIS ISF has a magnitude < 5 % (Fig. 12d-f). The linear correlations are as high as for the un-corrected case and the RMSD reduces to around 6 % (OSI-450, NT1-SSMI) and 7 % (CBT-SSMI) (compare to Fig. 12a-c

with Fig. 9b, e, h); the slope is almost identical with the identity line for OSI-450 and NT1-SSMI. We note that if the results of this bias correction prove to be of equal quality for other parts of the peak-melt period and other years, one could use the respective equations to obtain an independent estimate of the ISF from the entire PMW SIC data record, i.e. from 1979 to today. This could serve as an important boundary condition for the estimation of the surface albedo independent of daylight and cloud cover, complementing existing data sets and aiding in their evaluation (e.g. Riihela et al., 2010, 2017). For melt

advance (Fig. S5a-f), differences between bias-corrected PMW SIC and MODIS ISF are considerably larger than for peak melt, especially for CBT-SSMI and NT1-SSMI, and while the slopes all agree quite well with the identity line, RMSD values are much larger than for peak melt. This is also well evident from the larger scatter of value pairs in the respective 2D-histograms. During melt advance it appears advisable to use the non-truncated SIC values, because the fraction of SIC > 100 % is the highest during the summer melt cycle (see Sect. 5.1.1); at this stage we did, however, not further quantify the effect

this may have on the results of the bias correction performed.

As expected, the bias correction towards MODIS ISF has a large impact when the bias-corrected PMW SIC is now compared to MODIS SIC. The difference bias-corrected PMW SIC minus MODIS SIC is negative all over and takes a magnitude of about 25 %. We find a relatively homogeneous distribution of differences (Fig. 12j-l). We find value pairs in the respective 2D-histograms to be confined below the identity line around a linear regression line with a slope slightly larger than

1 and linear correlations comparable to the un-corrected PMW SIC (Fig. 6b, e, h). Most striking is the similarity of the distributions in the maps and 2D-histograms across the three products and the fact that the RMSD between bias-corrected PMW SIC and MODIS SIC not only agrees within 1 % among the three products tested but also agrees with the modal MPF of 25 % for DOY 201 of the year 2009 (see Fig. 2c). In contrast, during melt advance with melting and frozen, wet and dry surface co-existing the results of the bias correction of PMW SIC appear less convincing (Fig. S5g-l). Here CBT-SSMI

provides a difference bias-corrected PMW SIC minus MODIS SIC which in the central Arctic Ocean is uniform at about 10 %. This matches well with the MPF map and the first mode (9 %) of the bi-modal MPF distribution for DOY 169 of the year 2010 (Fig. 2b). The other differences range between -40 % and 10 % demonstrating that during melt advance a bias correction as proposed is potentially of limited value.

We summarize: in the short term, values held in Table 3 and Table 4 allow users of the existing PMW SIC datasets

to bias correct these products towards either true SIC or net ISF. As predicted by physics, the correction towards net ISF is more skilled than towards SIC. The bias correction towards ISF reconcile the various PMW SIC products in terms of true SIC. The bias correction towards ISF is however not necessarily useful in practice. Indeed, users must now rely on additional source of information to link their SIC (e.g. from a geo-physical model) to a measure of the ISF. This for example requires a trustworthy representation of the evolution of melt ponds on sea ice in their model. Several such melt pond schemes are being

developed (e.g. Pedersen et al., 2009; Flocco et al., 2010; Scott and Feltham, 2010; Holland et al., 2012; Skyllingstad et al.,





2015; Popović and Abbot, 2017) but their application and evaluation reveal some challenges remain (e.g. Light et al., 2015; Tsamados et al., 2015; Zhang et al., 2018; Burgard et al., 2019; Dorn et al., 2019). Still, in the long run, using PMW SIC as an observation of net ISF should be favoured, as it is more meaningful and will be more accurate. This will especially be the case when producers of PMW SIC datasets put additional effort in improving their algorithms and/or ice tie-point selection schemes

to actually retrieve un-biased observations of the net ISF. There is furthermore no doubt that both improving melt pond schemes in models and designing better PMW-based SIC algorithms in summer will benefit from better accuracy and availability of EO-based melt-pond fraction climate data records from visible / infrared imager instruments such as NASA MODIS (e.g., Rösel et al., 2012), the European Space Agency's MEdium Resolution Imager Sensor (MERIS) (e.g., Istomina et al., 2015; Zege et al., 2015), or the Copernicus Ocean and Land Colour Imager (OLCI). There is a critical EO-observation gap to be

filled here in order to further improve the Sea Ice ECV.

### 4.5 The impact on sea-ice area

Independent of the way forward and future attempts to get closer to what appears to be physically more correct when using satellite PMW data for SIC retrieval we note that there might be applications which require an accurate SIC *including* the melt ponds on top, i.e. without the need to understand why the PMW SIC does not match the actual net ISF. The classical

application would be the computation of the SIA which is the sum of the actually ice-covered area fraction of all grid cells or, in other words, the sum of the area of all ice-covered grid cells weighed by SIC. We demonstrated in Sect. 4.1 which groups of products over- and/or under-estimate MODIS SIC where and by which amount (Figs. 5, 6 and S1). We illustrated that on a pan-Arctic scale, averaged over years 2003-2011 group I products exhibit a near-0 % bias, while group III products (actually only NT1-SSMI, ASI-SSMI behaves like group I) appear to under-estimate MODIS SIC by 5-10 % during peak melt and end-

of-melt while group II products appear to over-estimate MODIS SIC by around 10 % (see Fig. 8). We illustrate in Fig. 13a that this statement holds for melt-pond fractions up to 30 %, for NT1-SSMI and group II even up to 40 %. In addition, Fig. 13b further illustrates how well the difference PMW SIC minus MODIS ISF can be seen as a linear function of the MPF for group I products – at least up to a MPF of ~ 30 %. This underlines both, applicability and limitation of the bias-correction introduced in Sect. 4.4 as a function of MPF.

Coming back to the computation of SIA and the potential influence of melt ponds: As shown in Kern et al. (2019) and Ivanova et al. (2014), the choice of the product for the computation of SIA from PMW SIC data makes a difference. For months July through September of years 2002-2011, the SIA computed from PMW SIC of group I products is ~400 000 km² larger than SIA computed from NT1-SSMI (group III) and ~600 000 km² smaller than SIA computed from group II products (Kern et al., 2019, Fig. G2g-i). The average pan-Arctic MODIS SIC for these months is between 85 % and 90 % (Fig. 7).

Considering a value of 90 % and assuming an extent of 6 million square kilometres to be covered by some amount of sea ice on average for these months, we end up with a SIA of about 5.4 million square kilometres based on MODIS. Group I products, exhibiting zero bias to MODIS SIC (Fig. 7a-d, Fig. 13a) yield the same SIA estimate. NT1-SSMI, exhibiting a negative bias of 5-10 % (Fig. 7i, Fig. 13a), say 7 %, i.e. a pan-Arctic average SIC of 83 %, yields a SIA of 5.0 million square kilometres. Group II products, exhibiting a positive bias of ~10 % (Fig. 7e-g, Fig. 13a), i.e. a pan-Arctic average SIC of 100 %, yield a

SIA of 6.0 million square kilometres. Based on these considerations we can conclude that the summer-time differences between the SIA estimates of the 10 products presented by Kern et al. (2019) can be explained well with the differences between PMW SIC and MODIS SIC presented in this paper.





## 5 Discussion and Conclusions

### 5.1 Discussion

#### *5.1.1 Using non-truncated sea-ice concentrations*

Our results presented in Sects. 3 and 4 are based on truncated sea-ice concentrations, i.e. SIC values set to exactly 100 % (exactly 0 %) in case the natural retrieval results in values > 100 % (< 0 %); see Kern et al. (2019) for more details. Of the product groups used (see Table 1), only group I offers non-truncated SIC values. We repeat our analyses with these non-truncated values and show selected results in Fig. 14 and Table 5. Figure 14 illustrates that a substantial fraction of the value

pairs of the 2-D histograms for the period 2003 to 2011 shown in the top row is located above the horizontal dotted line indicating 100 % PMW SIC – here for group I product OSI-450. Particularly during pre-melt (Fig. 14a, e) and melt advance (Fig. 14b, f) we find counts close to 1000 for PMW SIC values up to 105 % and 110 %, respectively; these are all set to 100 % in the panels of the bottom row in Fig. 14. The number of data points with PMW SIC > 100 % is much smaller for peak melt (Fig. 14c) but increases again for end-of-melt (Fig. 14d), albeit with considerably lower counts than during, e.g., melt

advance. The regression lines and statistical parameters given in the panels of Fig. 14 suggest: using non-truncated PMW SIC values has little impact on the statistical inter-comparison results during peak melt. However, there is a notable increase in the slope between PMW SIC and MODIS SIC during melt advance and, more importantly, there is a small increase in the linear correlation and a notable increase in the slope between PMW SIC and MODIS ISF (see Table 5). Because of this it could have been an advantage to carry out the bias correction described in Sect. 4.4 using the non-truncated instead of the truncated PMW

SIC data – especially for the case illustrating melt advance conditions (see Fig. S5). But at this stage further investigations into this topic are beyond the scope of this paper.

#### *5.1.2 Understanding our observations*

Our results demonstrate that the different products respond quite differently to the changes in the sea-ice cover during summer melt, and that none of them is doing things quite right. This is not surprising given the variety of different sea-ice and

snow physical properties relevant for satellite PMW sensing of sea ice – open and re-frozen melt ponds, slush, saturated or wet snow, new snow, coarse-grained melting or re-frozen snow, bare melting ice, bare dry ice, submerged ice and various forms of new ice – co-existing during summer at pan-Arctic scale but possibly even within one satellite footprint. These physical properties not only undergo substantial changes during the melt season, they also have a large spatiotemporal variability. The net surface energy balance driving the melting or freezing is very sensitive to variations in the cloud cover and

to precipitation events, which can vary on short temporal and local spatial scales. Melting and re-freezing of coarse-grained snow or formation of a thin ice cover at the melt-pond surface can occur within a few hours.

Besides melt ponds wet snow and melting and re-frozen coarse-grained snow are the most relevant surface parameters. At the microwave frequencies used in this paper the emissivity of the wet snow cover is close to 1, resulting in a microwave TB close to 273.15 K – the melting temperature of snow. Typical increases in microwave TB due to an increase in snow

wetness range between 10-15 K and 60 K (Kern et al., 2016, Table 1). The magnitude of this TB increase depends on the sea-ice emissivity being a function of frequency and polarization. The increase is higher for multiyear than first-year ice. It is higher at horizontal than vertical polarization and at higher (near-90 GHz) than lower (19 GHz) frequencies. This is all in accord with the lower TB of MY ice than FY ice and the lower TB at horizontal than vertical polarisation of winter sea ice. Concomitant is a decrease in the TB polarization difference, e.g. PR19 and PR89, as well as a decrease in the magnitude of

TB gradient ratios, e.g. GR3719 and GR8919, quantities that are used in the NT1-SSMI and NT2-AMSR-E algorithms (see Kern et al. (2019) for a summary of relevant technical aspects of the 10 algorithms used, and the definition of the PR and GR notations). Typical decreases in microwave TB due to an increase in snow grain size, e.g. due to surface refreezing or surface crust formation, are around 15-35 K (Kern et al., 2016, Tables 1 and 3). The magnitude of such a decrease is larger at horizontal





than vertical polarization, and larger at higher than lower frequencies. Concomitant is an increase in the magnitude of, e.g.,
PR19 and GR3719 by 0.02 and 0.05, respectively. Such increases correspond to about 10 % in SIC. Melting of a coarse-grained snow cover reverts the above-mentioned changes, causing diurnally changing microwave TB values should melt-refreeze cycles commence.

In summary, whenever the surface conditions become wetter, microwave TB increase, while polarization and frequency differences decrease. Whenever surface conditions become drier, microwave TB decrease, while polarization and
frequency differences increase. This view is certainly a simplification of the true conditions which are more complex due to the vertical structure of the snow cover, the different near-surface properties of first-year ice compared to multiyear ice, melt-pond drainage and other processes. However, this view allows us to understand that during summer, differences between an actually observed microwave TB or TB difference and an ice tie point can be caused by the mismatch between actual and tie-point conditions with respect to the representation of: i) melt ponds; ii) snow wetness; iii) snow grain size / surface type; iv)
ice type; v) a mixture of all these.

The implications for the SIC retrieval depend on the type and update interval of the so-called tie points of pure sea ice, i.e. 100 % sea-ice concentration (see e.g. Lavergne et al., 2019). The ASI algorithm (group III) uses one global fixed sea ice tie point value (Kaleschke et al., 2001). NT1-SSMI (group III) uses one fixed set of fixed TB values for first-year ice and multiyear ice. NT2-AMSR-E (group IV) uses sets of 12 fixed TB values of all involved channels (see Table 1) of three different
ice types: thin ice, ice type A (merges first-year and multiyear ice) and ice type C (sea ice with a thick snow cover); the number 12 results from the 12 different atmospheric states used to compute the look-up tables for the SIC retrieval (Markus and Cavalieri, 2009). All other products (groups I and II), except the contribution of NT1-SSMI to the NOAA CDR product, use an ice line which interpolates between signatures of first-year and multiyear ice and which is updated daily (Lavergne et al., 2019, Comiso and Nishio, 2008). For group I products this ice line is computed from TB measurements over closed ice within
a moving 15-days interval centred at the day of the actual SIC retrieval; closed ice is defined as grid cells with > 95% NASA-Team algorithm SIC. A post-processing step optimizes the location of the ice line with respect to the different TB values encountered as function of ice type. The Comiso bootstrap algorithm (CBT-SSMI and CBT-AMSR-E) derives the ice line via linear regression analysis of the respective TB value cluster. This is done in both TB spaces, i.e. TB37V / TB37H used for SIC > 90 % and TB37V / TB19V used for SIC ≤ 90 % (Comiso et al., 1997). The offset (or intercept) of the obtained linear
regression line is increased by a few Kelvin to account for the presence of some open water (2-3 %) in closed ice areas (Comiso and Nishio, 2008; Comiso, 2009). These differences in the ice tie points already suggest that the different products represent the actual sea-ice conditions with different levels of accuracy. None, to our best knowledge, of the algorithms used in the 10 products compared employ regionally varying ice tie points notwithstanding the large spatial variability of the relevant physical properties during the melt season.

*5.1.2.1 Example 1: pre-melt conditions*

For all groups we observe small areas of elevated positive differences PMW SIC minus MODIS ISF (Fig. 5a-d). These areas can be explained with the concurrent melt-pond fraction. An influence by elevated snow wetness is unlikely, because this would cause an increase in PMW SIC which in turn would result in an overestimation of both MODIS SIC *and* MODIS ISF. However, group I and III products reveal patches of MODIS SIC and ISF under-estimation (Fig. 5b, d, see also
Figs. S2 and S4); these are not observed for groups II and IV. As *one possibility* these patches could be explained by a re-frozen surface / coarse-grained snow not represented in the ice tie points. NT1-SSMI (group III) PMW SIC is based on PR19 and GR3719 and the above-mentioned surface conditions would cause an under-estimation of the SIC (see Kern et al., 2016, Fig. 6a: respective data pairs would move away from the red ice line towards the open water tie point). The algorithms of group I use NT1-SSMI SIC > 95 % as a-priori information for the computation of the ice tie point (Lavergne et al., 2019). Grid
cells with an actual near-100 % SIC, where such an under-estimation by NT1-SSMI occurs under the mentioned surface





conditions, are possibly excluded from the tie-point estimation. Therefore regions covered with near-100 % sea ice subject to such surface conditions might be excluded from the ice tie-point estimation. As a consequence such regions are not represented by the ice tie point, we have a mismatch between actual and tie-point conditions and the retrieved SIC is biased low.

### 5.1.2.2    Example 2: melt conditions

We observe areas with near-100 % MODIS SIC, near-0 % difference of PMW SIC to MODIS SIC, MODIS ISF over-estimation by 10-15 % and a MPF of 10-15 %, e.g. for group II, CBT-SSMI, central Arctic Ocean (Figs. 5f and 8f). One would expect that the open water of the non-zero MPF lowers the actually observed TB and that therefore the actual PMW SIC under-estimates MODIS SIC. This is not the case. We offer three explanations. Explanation A: the ice tie point includes some influence of melt ponds. In that case the ice tie point (see e.g. the ice line in Kern et al., 2016, Fig. 6c, d) would be located at
a lower TB value slightly closer to the open water tie point. The observed TB would then match with this ice tie point – provided that ice surface properties between the melt ponds match. The retrieved SIC would be close to 100 %. We hypothesize that this is one of the most likely reasons for over-estimation of MODIS ISF by group I products. These products use NT1-SSMI SIC > 95% to define regions for ice-tie point retrieval (Lavergne et al., 2019), regions which according to the results of our paper exhibit a non-zero melt-pond fraction. Explanation B: the surface between the melt ponds is wet but this is not
represented by the ice tie point. In that case the observed TB is – on the one hand – lowered by the melt ponds but – on the other hand – increased by the wet snow. Both effects could compensate such that the observed TB is close to the ice tie point yielding near-100 % SIC. Evidence for an increase in TB during summer melt in June is given, e.g. in Kern et al. (2016, Fig. 8a-c); the cluster of increased TB values is located considerably above the ice line concomitant with near-100 % MODIS ISF (Kern et al., 2016, Fig. 6c). Explanation C: the ice tie point represents a refrozen surface or multiyear ice and because of this
is located similarly closer to the open water tie point as in explanation (A). The observed TB would match the ice tie point for the wrong reason and the algorithm would provide near-100 % SIC.

In addition, we observe areas with 100 % PMW SIC coinciding with 85% for MODIS SIC and 25-30% for MPF, e.g. for group II, CBT-SSMI, Chukchi Sea (Figs. 5f and 8f). Here, despite a large open water fraction of 40-45 % is present, PMW SIC is 100 % and with that MODIS SIC (MODIS ISF) over-estimated by 15 % (40-45 %). All explanations suggested in the
previous paragraph might apply here – very likely in combination with each other. Such a large over-estimation of MODIS ISF would, if we use only explanation B, require TB values stemming from un-physically large sea-ice surface emissivities > 1 (not shown). Kern et al. (2016) computed the SIC using summer-time elevated microwave TB values for different algorithms and suggested that – theoretically – SIC values would need to be as high as 140 % for the fraction of the grid cell not covered by water to explain the observed differences between PMW SIC and MODIS ISF for MODIS SIC values > 90 %. In light of
the way ice tie points are derived in the Comiso bootstrap algorithm, a combination of i) inclusion of melt ponds in the tie point, ii) unaccounted wet snow / wet surface between the melt ponds and iv) ice type mismatches seems to be the most likely combination leading to the observed over-estimation.

### 5.1.2.3    Summary

The co-existence of different surface properties during summer adds complexity to the SIC retrieval using satellite
PMW TB observations. Our attempts to explain the observations suggest that an adequate understanding of – on the one hand – the actually encountered sea-ice and snow properties and – on the other hand – the properties represented by the ice tie points is required. The influence different surface properties exert on the actually measured TB or TB differences like PR19 or GR3719 can cancel out. Examples of such properties are the co-existence of melting and re-frozen coarse-grained snow or the co-existence of wet snow and melt ponds. A consequence of this is that despite the actual surface conditions do not match the
ice tie-point conditions, retrieved PMW SIC appear to be accurate. In addition, it needs to be better understood how ice tie



points are derived during summer conditions and how their validity can be assessed as a function of location and time. Based on our findings, one of the largest issues could be the inclusion of an unknown amount of melt ponds into the ice tie point.

## 5.2    Conclusions

Following up on the release of three new global sea-ice concentration (SIC) climate data records (CDRs) described in Lavergne et al. (2019), this paper focuses on an inter-comparison of these three CDRs and seven other SIC products (see Kern et al., 2019) with estimates of the SIC and the net ice-surface fraction (ISF) in the Arctic during summer (May through September) of years 2003 through 2011 obtained from satellite observations by the MODIS sensor. The motivation for this dedicated paper is the fact that it is particularly challenging to derive the SIC of melting sea ice. It is impossible with the current microwave radiometer sensors to distinguish water in melt ponds on top of the sea ice from the water in the leads between the sea-ice floes. What we expect to measure with the microwave radiometer sensor is therefore the ISF even though this may not be what scientists normally understand by the term "sea ice concentration". The ISF radiometric signature and especially its variability is difficult to characterise when the sea ice is melting and this results in large random and systematic uncertainties when retrieving the SIC and ISF from PMW observations. Our study employs 10 SIC products which we assign to four groups based on their retrieval algorithm (see Table 1 and Kern et al., 2019).

Overall we find group I products (SICCI and OSI-450, see Table 1) exhibit a near-0 % bias to the MODIS Arctic average SIC – independent of melt-pond fractions up to ~35 % (Fig. 13a). Group II (CBT-SSMI, CBT-AMSRE and NOAA-CDR) and IV (NT2-AMSRE) products have a positive bias of 5-10 % and NT1-SSMI (group III) has a negative bias of 5-10 %; ASI-SSMI (group III) is similar to group I. However, these small overall biases are the result of wide-spread, spatiotemporally varying positive and negative differences of substantial magnitude which cancel out in a pan-Arctic mean for some of the products. Magnitudes of these biases frequently reach up to 20-25 % for groups I and III and up to 30-35 % for groups II and IV.

By comparing PMW SIC with MODIS ISF and the MODIS melt-pond fraction (MPF) we find that all 10 SIC products substantially over-estimate MODIS ISF. This over-estimation is generally related to the MPF but the degree of over-estimation varies between groups of products. On pan-Arctic scale, group I products over-estimate MODIS ISF by almost exactly the overall mean MPF for values below ~30 % (Fig. 13 b). In contrast, group II and IV products over-estimate MODIS ISF by an amount 5-10 % higher than the mean MPF while NT1-SSMI (group III) over-estimate MODIS ISF by an amount 5-10 % smaller than the mean MPF.

The observed differences between PMW SIC and MODIS SIC or ISF cannot be explained by the presence of melt ponds alone. Regions exhibiting the highest MPF should coincide with regions of the largest over-estimation of MODIS ISF by PMW SIC. This often is not the case. For example, we find regions with near-100 % PMW SIC coinciding with MODIS ISF and MODIS SIC of 55 % and 85 %, respectively. The associated MPF of ~30 % is 15 % smaller than the amount MODIS ISF is over-estimated by PMW SIC. By taking into account the different PMW data used by the various algorithms and relevant surface properties other than melt ponds – such as wet snow, (refrozen) coarse grained snow or a frozen surface in general – we discuss potential reasons for our observations. For the near-19 GHz and near-37 GHz frequency channels often used for SIC retrieval (see Table 1), open water in the form of melt ponds reduces the observed PMW brightness temperatures. However, during summer the surface / snow between melt ponds is likely wet which results in an increase of their brightness temperatures compared to typical winter and spring conditions. The two effects counteract and might result in a PMW SIC close to 100 % despite a considerably smaller ISF of, e.g., 80 % caused by the melt ponds. This applies in particular when the ice tie point does not well represent wet snow conditions. Another scenario which would have the same effect is that the ice tie point does represent wet snow conditions between the melt ponds well but does *in addition* include an un-known amount of open water due to melt ponds, i.e. does not represent 100 % melt-pond free sea ice. Our results illustrate that of the two



groups of products employing advanced ice tie point retrieval methods, i.e. groups I and II, the methods of group I are considerably more successful in mitigating the unwanted influence of surface properties other than melt ponds.

Our inter-comparison reveals similarity in the results between group I and III products on the one hand and between group II and IV products on the other hand. This similarity is particularly interesting because ice tie-point estimation differs substantially between group I and III, and between group II and IV. While groups I and II have comparably advanced schemes to derive the ice tie point at daily temporal scale, ice tie points are fixed throughout the year for algorithms of the other two groups. This appears to call for a revision of the currently used concepts to derive and use ice tie points during summer. One potential solution to reduce SIC biases reported in this paper would be to add regional variation by, e.g. incorporating the

different regionally varying stages of melt into the ice tie-point estimation. This could be done, e.g., by using maps of melt onset derived from PMW observations (e.g. Stroeve et al., 2014; Markus et al., 2009) – possibly in combination with atmospheric re-analysis data or observations of the ice surface temperature. Another solution could be to simplify the entire SIC retrieval process by, e.g. assuming melt conditions globally and derive a global ice tie point for melting conditions. The smaller sensitivity of lower frequency channels to changes in snow grain size or snow wetness makes these particularly good

candidates for ice tie-point optimization during summer conditions. Such channels are for example offered by AMSR-E, AMSR2, the Soil Moisture and Ocean Salinity (SMOS) mission and the planned Copernicus Imaging Microwave Radiometer (CIMR) mission, i.e. near 7 GHz or even near 1 GHz.

One goal of such an optimization could be to further support the switch from sea-ice extent (SIE) to sea-ice area (SIA) as the main parameter to monitor long-term changes of the Arctic sea-ice cover. Sea-ice covers sharing the same sea-ice edge

provide the same SIE regardless of how open the sea-ice cover inside the ice edge actually is while SIA allows one to distinguish between a comparably open sea-ice cover = low SIA and a highly compact sea-ice cover = large SIA. Hence the SIA clearly outperforms SIE in terms of providing information about the status of the sea-ice cover inside the ice edge (e.g. Notz, 2014), but is also much more affected by systematic biases as documented here for all 10 algorithms during summer. Another goal could be to support moving away from retrieving sea-ice concentrations during winter and some highly

inaccurate, ill-defined quantity during summer, still called sea-ice concentration, and rather retrieve net ice-surface fraction year-round – the quantity which by physical means it the one accessible with microwave radiometry year-round. Such a switch will require improved algorithms for PMW observations, improved melt-pond formulations in geophysical models, and better and accessible melt-pond fraction datasets. We recommend that the sea-ice communities work towards such a switch to improve sea ice as an Essential Climate Variable (ECV).

*Data availability.* All sea-ice concentration products except SICCI-12km are publicly available from the sources provided in the reference list or in Kern et al. (2019). The SICCI-12km product is available upon request from T. Lavergne. The MODIS data set of sea-ice concentration, melt-pond fractions and net ice surface fraction is available from: https://doi.org/10.1594/WDCC/MODIS__Arctic__MPF_V02 , the standardized ship-based observations are available from: https://doi.org/10.26050/WDCC/ESACCIPSMVSBSIO .

*Author contributions.* SK wrote the manuscript. All co-authors contributed to the concept and work presented in the paper and also assisted in the writing. SK performed the data analysis and inter-comparison with contributions in the interpretation of the results from all co-authors.

*Competing interests.* The authors declare that they have no conflict of interest.

*Acknowledgements.* The work presented here was funded by EUMETSAT (through the 2nd Continuous Developments and Operation Phase of OSI SAF) and ESA (through the Climate Change Initiative Sea_Ice_cci project), and the German Research Foundation (DFG) Excellence Initiative CLISAP under Grant EXC 177/2. The publication itself is funded by the Deutsche



Forschungsgemeinschaft (DFG, German Research Foundation) under Germany's Excellence Strategy – EXC 2037 'CLICCS – Climate, Climatic Change, and Society' – Project Number: 390683824, contribution to the Center for Earth System Research and Sustainability (CEN) of the University of Hamburg.

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





## 7    Tables

**Table 1.** Overview of the investigated sea-ice concentration products. Column "ID (Algorithm)" holds to the identifier we use henceforth to refer to the data product, and which algorithm it uses. Group is an identifier for the algorithm concept used. Column "Input data" refers to the input satellite data for the data set. Columns "Tie points" and "Tie-point Update" refers to the type of tie points used and their update interval (see text for further details).

| ID (algorithm) | Group | Input data & frequencies | Grid resolution & type | Tie points | Tie-point update | Reference |
|---|---|---|---|---|---|---|
| OSI-450 (SICCI2) | I | SMMR, SSM/I, SSMIS 19.35 & 37.0 GHz | 25 km x 25 km EASE2.0 | Open water, ice line | Daily | Tonboe et al., 2016; Lavergne et al., 2019 |
| SICCI-12km (SICCI2) | I | AMSR-E, AMSR2 18.7 & 89.0 GHz | 12.5 km x 12.5 km EASE2.0 | Open water, ice line | Daily | Lavergne et al., 2019 |
| SICCI-25km (SICCI2) | I | AMSR-E, AMSR2 18.7 & 36.5 GHz | 25 km x 25 km EASE2.0 | Open water, ice line | Daily | Lavergne et al., 2019 |
| SICCI-50km (SICCI2) | I | AMSR-E, AMSR2 6.9 & 36.5 GHz | 50 km x 50 km EASE2.0 | Open water, ice line | Daily | Lavergne et al., 2019 |
| CBT-SSMI (Comiso bootstrap) | II | SMMR, SSM/I, SSMIS 19.35 & 37.0 GHz | 25 km x 25 km PolarStereo | Open water, ice line | Daily | Comiso, 1986; Comiso et al., 1997; Comiso and Nishio, 2008 |
| NOAA-CDR (NASA Team & Comiso bootstrap) | II | SSM/I, SSMIS 19.35 & 37.0 GHz | 25 km x 25 km PolarStereo | Open water, ice line & Open water, first-year ice, multiyear ice | Daily & fixed | Peng et al., 2013; Meier and Windnagel, 2018 |
| CBT-AMSR-E (Comiso bootstrap) | II | AMSR-E 18.7 & 36.5 GHz | 25 km x 25 km PolarStereo | Open water, ice line | Daily | Comiso et al., 2003; Comiso and Nishio, 2008; Comiso, 2009 |
| ASI-SSMI (ASI) | III | SSM/I, SSMIS 85.5 GHz | 12.5 km x 12.5 km PolarStereo | Open water, sea ice | Fixed | Kaleschke et al., 2001; Ezraty et al., 2007 |
| NT1-SSMI (NASA-Team) | III | SMMR, SSM/I, SSMIS 19.35 & 37.0 GHz | 25 km x 25 km PolarStereo | Open water, first-year ice, multiyear ice | Fixed | Cavalieri et al, 1984; 1992; 1999 |
| NT2-AMSR-E (NASA-Team-2) | IV | AMSR-E 18.7, 36.5 & 89.0 GHz | 25 km x 25 km PolarStereo | Open water, thin ice, ice type A, ice type C | Daily* | Markus and Cavalieri, 2000; 2009 |

**Table 2.** Overall mean difference: individual algorithm SIC minus ensemble mean SIC in percent ice concentration for the Arctic for winter (months January and February) and summer (months July (see Figure 4) and August). N denotes the total number of valid data pairs with SIC > 15.0% (see also Kern et al., 2019).

| | | Group I | | | | Group II | | | Group III | | IV |
|---|---|---|---|---|---|---|---|---|---|---|---|
| | N | SICCI12 | SICCI25 | SICCI50 | OSI450 | CBT-SSMI | NOAA-CDR | CBT-AMSRE | ASI-SSMI | NT1-SSMI | NT2-AMSRE |
| Jan/Feb | 9821 | -1.3 | -1.0 | -1.2 | -0.6 | +2.2 | +2.2 | +2.5 | -2.8 | -2.7 | +2.7 |
| July/Aug | 5698 | -2.7 | -3.6 | -4.0 | -3.0 | +5.6 | +5.9 | +5.8 | +1.2 | -8.1 | +3.0 |

**Table 3.** Average values of linear correlation, root-mean-squared difference (RMSD), and slope as well as intercept of the linear regression between passive microwave and MODIS sea-ice concentration for product groups I to IV (see text for further information). The averages are derived as the arithmetic mean from all 8-day period values of products within one group falling into "pre-melt": DOY 129, 137, and 145, "melt advance": DOY 153 to 185, "peak melt": DOY 193 to 233, and "end-of-melt": DOY 241 and 249.

| parameter | correlation | | | | RMSD [%] | | | | slope | | | | intercept [%] | | | |
|---|---|---|---|---|---|---|---|---|---|---|---|---|---|---|---|---|
| group | I | II | III | IV | I | II | III | IV | I | II | III | IV | I | II | III | IV |
| pre-melt | 0.39 | 0.54 | 0.53 | 0.47 | 5.8 | 4.5 | 5.6 | 4.6 | 0.77 | 0.74 | 1.05 | 0.71 | 21.8 | 27.4 | -6.5 | 29.9 |
| melt advance | 0.65 | 0.61 | 0.62 | 0.61 | 7.4 | 7.3 | 8.2 | 6.5 | 1.03 | 0.75 | 0.99 | 0.78 | -5.4 | 26.0 | -2.8 | 22.1 |
| peak melt | 0.82 | 0.80 | 0.83 | 0.74 | 8.1 | 11.4 | 9.2 | 10.1 | 1.27 | 1.03 | 1.26 | 1.09 | -23.5 | 1.2 | -23.8 | -2.3 |
| end of melt | 0.78 | 0.75 | 0.81 | 0.61 | 9.5 | 11.7 | 12.0 | 11.7 | 1.16 | 1.05 | 1.31 | 0.84 | -16.3 | 2.8 | -30.7 | 18.1 |





**Table 4.** Mean values of linear correlation, root-mean-squared difference (RMSD), and slope as well as intercept of the linear regression between passive microwave sea-ice concentration and MODIS ice-surface fraction for product groups I to IV (see text and caption of Table 3 for further information).

| parameter | correlation | | | | RMSD [%] | | | | slope | | | | intercept [%] | | | |
|---|---|---|---|---|---|---|---|---|---|---|---|---|---|---|---|---|
| group | I | II | III | IV | I | II | III | IV | I | II | III | IV | I | II | III | IV |
| pre-melt | 0.38 | 0.52 | 0.51 | 0.48 | 7.9 | 7.9 | 7.1 | 7.9 | 0.38 | 0.37 | 0.54 | 0.39 | 59.9 | 64.8 | 44.0 | 62.0 |
| melt advance | 0.72 | 0.62 | 0.70 | 0.62 | 15.8 | 21.1 | 15.9 | 20.0 | 0.59 | 0.40 | 0.58 | 0.43 | 47.0 | 65.1 | 44.7 | 61.2 |
| peak melt | 0.80 | 0.75 | 0.81 | 0.73 | 24.5 | 33.0 | 23.2 | 30.5 | 0.86 | 0.71 | 0.85 | 0.74 | 30.3 | 48.6 | 29.4 | 43.6 |
| end of melt | 0.75 | 0.69 | 0.78 | 0.59 | 18.3 | 26.5 | 17.8 | 25.4 | 0.69 | 0.59 | 0.77 | 0.49 | 34.4 | 50.5 | 26.1 | 55.4 |


**Table 5.** Linear correlation and slope of the linear regression between group I OSI-450 sea-ice concentration and MODIS SIC (top part) or MODIS ISF (bottom part) for years 2003-2011 illustrating the difference between using truncated or non-truncated PMW SIC values in the 2-D histograms of the kind shown in Figs. S1 and S3. For pre melt and peak melt only the two 8-day periods directly next to the melt-advance period are shown. Numbers in bold font denote a correlation being larger by 0.02 and a slope being larger by 0.5 because of using

non-truncated instead of truncated PMW SIC values.

| | | Pre-melt | | Melt advance | | | | | Peak melt | |
|---|---|---|---|---|---|---|---|---|---|---|
| MODIS SIC | 8-day period | 137 | 145 | 153 | 161 | 169 | 177 | 185 | 193 | 201 |
| Correlation | truncated | 0.44 | 0.49 | 0.56 | 0.70 | 0.66 | 0.67 | 0.73 | 0.82 | 0.86 |
| | non-truncated | 0.44 | 0.48 | 0.56 | 0.70 | 0.67 | 0.68 | 0.73 | 0.82 | 0.86 |
| Slope | truncated | 0.913 | 1.022 | 1.117 | 1.331 | 1.185 | 0.935 | 0.943 | 1.144 | 1.269 |
| | non-truncated | 0.945 | 1.045 | 1.145 | **1.407** | **1.294** | **1.001** | 0.959 | 1.154 | 1.277 |
| MODIS ISF | | | | | | | | | | |
| Correlation | truncated | 0.43 | 0.48 | 0.58 | 0.78 | 0.78 | 0.79 | 0.76 | 0.82 | 0.86 |
| | non-truncated | 0.41 | 0.47 | 0.58 | 0.78 | **0.80** | **0.81** | 0.77 | 0.83 | 0.86 |
| Slope | truncated | 0.467 | 0.466 | 0.510 | 0.679 | 0.676 | 0.621 | 0.670 | 0.801 | 0.891 |
| | non-truncated | 0.468 | 0.469 | 0.519 | 0.718 | **0.743** | **0.678** | 0.689 | 0.810 | 0.899 |

## 8 Figures

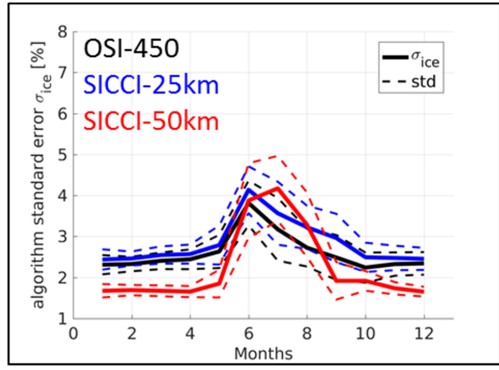

**Figure 1.** Seasonal cycle of the multi-annual (2002-2011) average sea-ice concentration algorithm standard error for the Arctic for grid cells

with > 90 % sea-ice concentration for OSI-450, SICCI-25km and SICCI-50km products. Shown are the mean (solid line) and its standard deviation (dashed line denoted "std").







**Figure 2.** Sample maps of the Arctic melt-pond fraction from the MODIS data set for a) day-of-year (DOY) 129 (May 9-16) 2003, b) DOY 169 (June 18-25) 2010, c) DOY 201 (July 20-27) 2009, and d) DOY 241 (Aug. 29-Sep. 5) 2006, illustrating the conditions during pre-melt, melt advance, peak melt, and end of melt, respectively. Black color denotes open water, missing (note in this context the curvilinear one-grid-cell wide features with missing data which originate from the gridding process) or invalid data and clouds. Melt-pond fractions < 5 % are displayed white. The histograms show the distribution of the melt-pond fraction for above-quoted DOY for every year of the period 2003-2011.



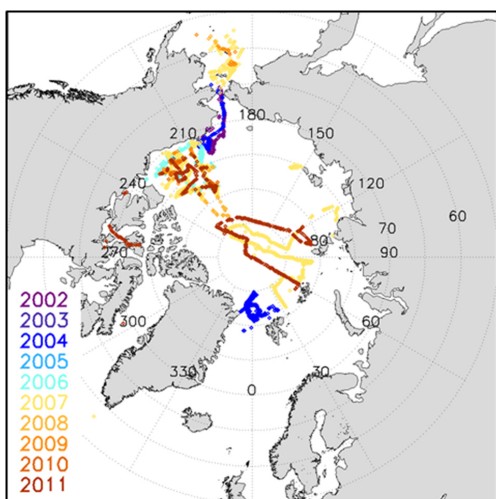

**Figure 3.** Spatiotemporal distribution of ship tracks in the Arctic from which ship-based visual observations of the sea-ice cover were used.



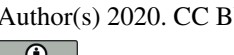


**Figure 4. (a)** to **(j)** Maps of the difference between the multi-annual average monthly SIC of the individual algorithms and the 10-algorithm ensemble median multi-annual average monthly SIC **(k)** for the Arctic for July 2003-2011. Differences are only computed for sea-ice concentration of both data sets > 15%. Roman numbers in bold font denote the group (see Table 1) to which the algorithm is assigned.




**Figure 5.** Maps of the difference PMW minus MODIS sea-ice concentration for a) to d): day-of-year (DOY) 129 (May 9-16) 2003, e) to h): DOY 169 (June 18-25) 2010, i) to l): DOY 201 (July 20-27) 2009, and m) to p): DOY 241 (Aug. 29-Sep. 5) 2006; these are the same periods as used in Fig. 2. The leftmost column shows OSI-450, representing group I, the second column CBT-SSMI, representing group II, the third column NT1-SSMI, representing group III, and the rightmost column NT2-AMSRE, representing group IV. Black areas denote invalid or missing data, clouds, or grid cells being ice-covered but not considered further in the analysis, e.g. in the Greenland Sea or Hudson Bay. Row starting with (a) are representative of pre-melt, row starting with (e) is melt advance, row starting with (i) is at the peak of melt, and row starting with (m) is at the end-of-melt.

**Figure 6.** Two-dimensional histograms of the distribution of PMW (y-axis) versus MODIS (x-axis) SIC data pairs using a bin size of 1 %
for the same 8-day periods as shown in Fig. 5e-p, i.e. melt advance, peak melt, and end of melt. The topmost row shows OSI-450 (for group
I), the second row CBT-SSMI (for group II), the third row NT1-SSMI (for group III), and the bottommost row NT2-AMSRE (group IV).
The thin black line is the identity line. The thick black line denotes the linear regression through the data pairs. At the top left of every image
we display the linear correlation coefficient R, the number of data pairs N and the root mean squared difference RMSD; the latter is given
in percent. The leftmost, middle and rightmost columns are representing melt advance, peak of melt, and end of the melt, respectively.
Respective scatterplots for pre-melt are shown in Figure S2.

**Figure 7.** Average seasonal cycle of the mean (limited to Arctic Ocean and Canadian Arctic Archipelago) MODIS SIC (in blue), PMW SIC (in red), their difference PMW minus MODIS SIC (in orange), and the MODIS melt-pond fraction (in cyan), averaged for each 8-day period for the years 2003-2011. Error bars denote one standard deviation of the mean. Roman numbers in bold font denote the group (see Table 1) to which the algorithm is assigned.





**Figure 8.** Maps of the difference PMW sea-ice concentration minus MODIS ice surface fraction (ISF) for the same 8-day periods as shown in Fig. 5. The leftmost column shows OSI-450, representing group I, the second column CBT-SSMI, representing group II, the third column NT1-SSMI, representing group III, and the rightmost column NT2-AMSRE, representing group IV. Black areas denote invalid or missing data, clouds, or grid cells being ice-covered but not considered further in the analysis, e.g. in the Greenland Sea or Hudson Bay. Row starting with (a) are representative of pre-melt, row starting with (e) is melt advance, row starting with (i) is at the peak of melt, and row starting with (m) is at the end of the melt.





**Figure 9.** Two-dimensional histograms of the distribution of PMW SIC (y-axis) versus MODIS ISF (x-axis) data pairs using a bin size of 1 % for the same 8-day periods as shown in Fig. 8e-p, i.e. melt advance, peak melt, and end of melt. The topmost row shows OSI-450 (for group I), the second row CBT-SSMI (for group II), the third row NT1-SSMI (for group III), and the bottommost row NT2-AMSRE (group IV). The thin black line is the identity line. The thick black line denotes the linear regression through the data pairs. At the top left of every image we display the linear correlation coefficient R, the number of data pairs N and the root mean squared difference RMSD; the latter is given in percent. The leftmost, middle and rightmost columns are representing melt advance, peak of melt, and end of the melt, respectively. Respective scatterplots for pre-melt are shown in Figure S4.


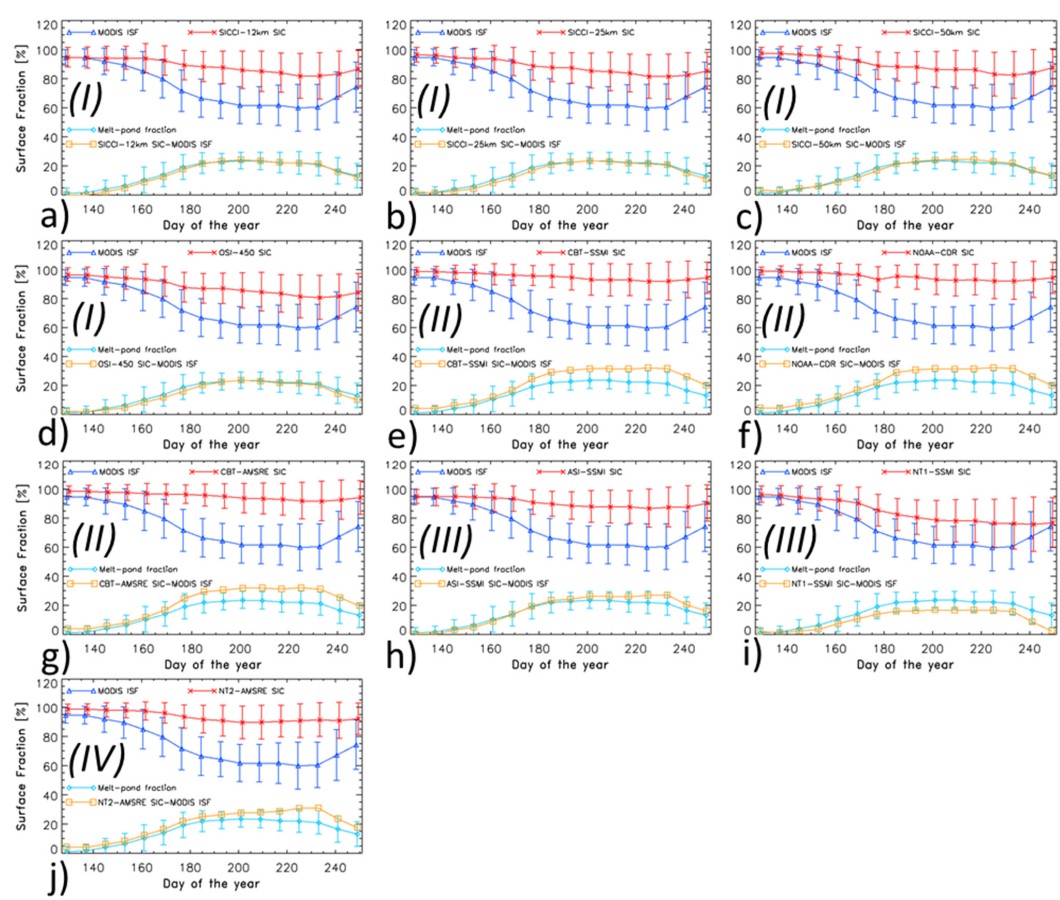

**Figure 10.** Average seasonal cycle of the mean (limited to Arctic Ocean and Canadian Arctic Archipelago) MODIS ISF (in blue), PMW SIC (in red), their difference PMW SIC minus MODIS ISF (in orange), and the MODIS melt-pond fraction (in cyan), averaged for each 8-day period over the years 2003-2011. Error bars denote one standard deviation of the mean. Roman numbers in bold font denote the group (see Table 1) to which the algorithm is assigned.





**Figure 11.** Scatterplots of co-located daily average SIC (black dots) and average ice-surface fraction (red dots) from visual ship-based observations (ASPeCt, x-axis) and the ten satellite SIC algorithm products (SAT, y-axes) for the Arctic for May through September for years 2002-2011 (see Fig. 3 for locations). Error bars denote one standard deviation of the average. Dotted lines denote the identity line. Solid lines denote the linear regression of the SIC data pairs. The mean difference (standard deviation), the linear regression equation, the squared linear correlation coefficient ($R^2$) and the number of valid data pairs N is given in the top left of every image for the daily SIC value data pairs. Red triangles (blue squares) denote the average SAT SIC at 10 % wide ship-based SIC bins (average ship-based SIC at 10 % wide SAT SIC bins). Roman numbers in bold font denote the group (see Table 1) to which the algorithm is assigned.

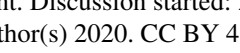



**Figure 12.** Illustration of the effect of a simple linear bias correction for an 8-day period during peak-melt (DOY 201, July 20-27, 2009).

Topmost row, panels a) to c): Two-dimensional histograms of the distribution of bias-corrected PMW SIC (y-axis) versus MODIS ISF (x-axis) data pairs. Second row, panels d) to f): Respective maps of the difference of bias-corrected PMW SIC minus MODIS ISF. Third row, panels g) to i) Two-dimensional histograms of the distribution of bias-corrected PMW SIC (y-axis) versus MODIS SIC (x-axis) data pairs. Bottommost row, panels j) to l): Respective maps of the difference bias-corrected PMW SIC minus MODIS SIC. Leftmost, middle and rightmost columns show OSI-450 (for group I), CBT-SSMI (for group II), and NT1-SSMI (for group III). Bin size in the histograms is 1 %. The quantities given in the top left corner are R: linear correlation coefficient, N: number of valid data pairs, and RMSD: root mean squared difference. The thin black line is the identity line; the thick black line denotes the linear regression through the data pairs.

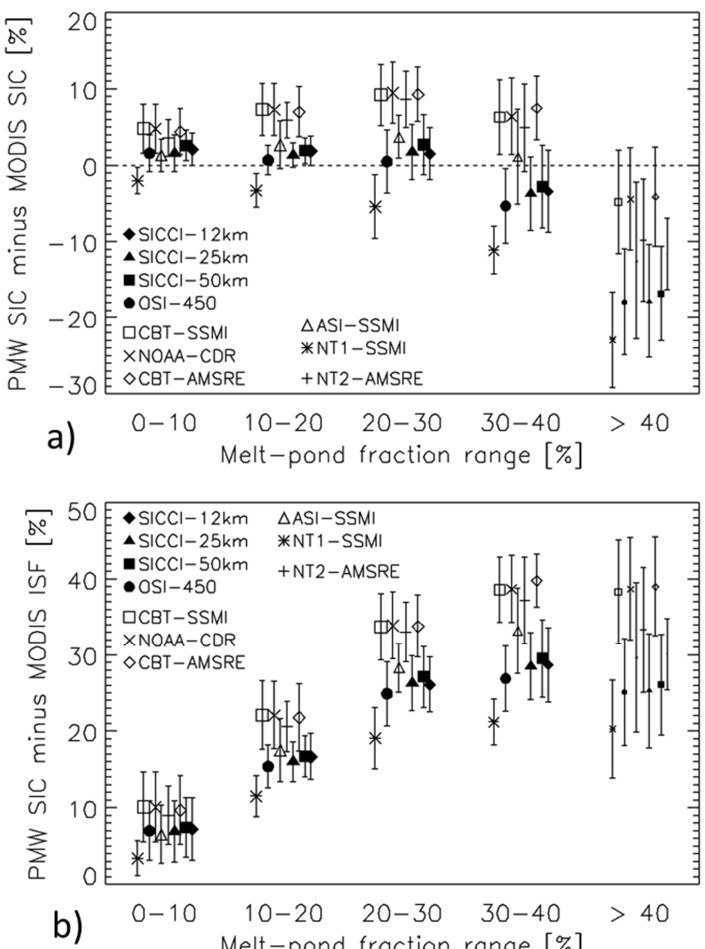

**Figure 13.** Mean difference PMW SIC minus MODIS SIC (image a)) and PMW SIC minus MODIS ISF (image b)) derived for all 8-day periods of the years 2003-2011 for all ten products separately for melt-pond fraction ranges 0-10 % to > 40 %. Error bars denote one standard deviation of the mean. Symbol size scales with the number of valid data pairs. The topmost four and bottommost three entries in left column of annotations denote group I (filled symbols) and group II, respectively. The topmost two entries and the last entry in right column of annotations denote group III and group IV, respectively.



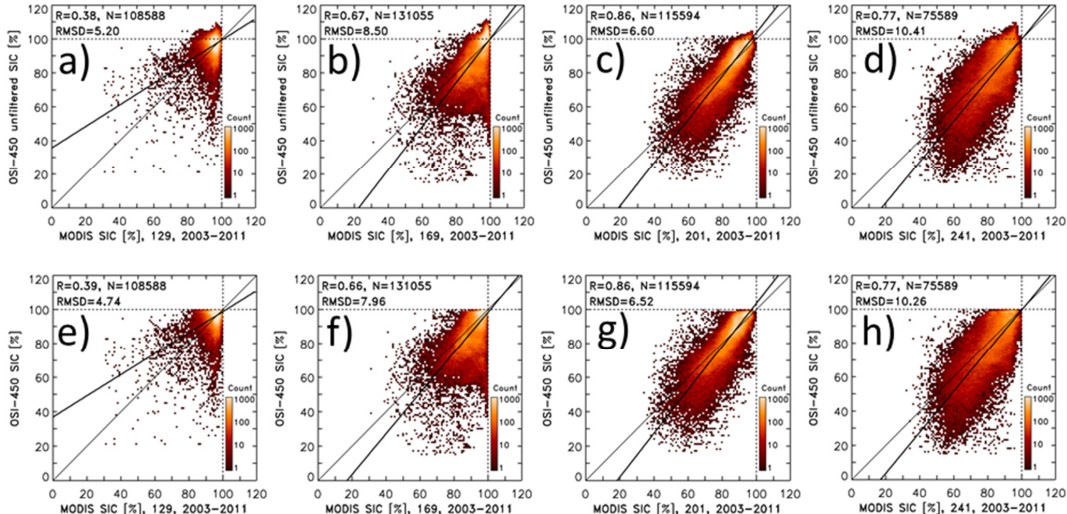

**Figure 14.** Illustration of the impact using the non-truncated (top row, panel a) to d) instead of the truncated (bottom row, panel e) to h) SIC data, for example OSI-450 (representative of group I products), when comparing PMW SIC with MODIS SIC. Shown are the two-dimensional histograms for all four 8-day periods starting at DOY 129, 169, 201, and 241 of the years 2003 through 2011 (see Fig. S1). The quantities given in the top left corner are R: linear correlation coefficient, N: number of valid data pairs, and RMSD: root mean squared difference. The thin black line is the identity line; the thick black line denotes the linear regression through the data pairs.