# Peer review of "Satellite Passive Microwave Sea-Ice Concentration Data Set Intercomparison for Arctic Summer Conditions"

_The Cryosphere, 2020_

## Referee Comment (RC1) · Anonymous Referee #1 · 25 Mar 2020

Dear authors of the manuscript tc-2020-35,

The manuscript is interesting and useful for the sea ice research. It is very useful to make this kind of comparisons. Comparisons to MODIS SIC give an idea of the weaknesses of the existing algorithms and also gives tools to correct or improve the current algorithms, especially in the summer conditions with the highest uncertainties. The suggested bias correction looks like a promising approach to get more accurate SIC or ISF estimates. Also the potential explanations for the algorithm behaviour in summer conditions are interesting and useful information and give good information for further research and algorithm development.

I only have a few minor comments:

1) The algorithms have been divided into four groups. I think the division has been in more detail reasoned in the previous TC paper of the authors. However, I would miss a short description/reasoning also here pointing out what are the actual differences between the algorithm groups and what are the similarities (and differences) within groups. I think this can be seen as a clustering analysis i.e. looking for an optimal set of clusters simultaneously minimizing within-cluster distance and maximizing between-cluster difference for a set of (selected) features.

2) There are formulas of the parameters discussed given. Even though they are mostly quite simple, formulas would make the presentation even more clear, e.g. SIA = \intergral SIC dA or something similar, now they have (only) been described in words.

3) Some abbreviations are not explained, at least PR and GR should be explained as they appear for the first time (even though they are clear for the most readers). Still check all of these. Possibly even (general) formulas for PR and GR could be given? Also open OSI SAF and SICCI.

4) Consider of replacing "<" and ">" in the text by "less than" and "greater/more than"

5) P2 L59: "...early in the melt season or on land-fast sea ice..." When this may occur over land-fast ice? Also during early melt season?

6) P3 L125: "... with the size of the field-of-view of..." Could this be e.g. diameter (I think the unit of the size should be km2)?

7) P7 L1149-150: "last accessed October 12, 2016". This is almost 4 years ago, update this.

8) P5 L187-189: "...excluded all the samples..." "...larger than 1". The ratio mean(MPF)/std(MPF) is the signal to noise ratio, if You exlude the values with high SNR then You only include the uncertain data? Or did You mean the coefficient of variation std(MPF)/mean(MPF) instead?

[Figure]

9) Conversion to Cartesian coordinates is mentioned. But is there a certain projection You are using. What are the units in the coordinate system (e.g. meters in which projection)? This is not very clear to me based on the manuscript.

10) You use the term Day of the Year, it is also often referred as the Julian day. I do not know which one is better in scientific articles, I have seen both practices.

11) P13 L534: "values held in Table 3...". Probably "values given in Tables 3 and 4" or "values held by tables 3 and 4" would sound better?

12) P15 L613: "accord" -> "accordance"?

13) P17 L697: "The influence different surface properties exert on the" -> "The influence exerted bydifferent surface properties..."?

14) Possibly You could also mention the bias correction approach of section 4.4. also in the final conclusion section (5.5.).

Sincerely,

———————————————————

---

## Referee Comment (RC2) · Anonymous Referee #2 · 8 Apr 2020

Summary

This paper compares ten passive microwave (PM) sea ice concentration algorithms during summer melt conditions in the Arctic. The comparisons are done relative to a MODIS-derived surface classification product that distinguishes open water, melt ponds, and unponded sea ice. Ship observations are also used in the comparison. The ten algorithms are split into four characteristic groups based on the formulation of the algorithms. Comparisons are made for sea ice concentration (SIC) and net ice-surface fraction (ISF = SIC minus melt pond fraction). The results show varied performance of the algorithms during different stages of the melt period. Summer SIC is overestimated

by some algorithms, but underestimated by others. However, ISF is systematically overestimated by all algorithms. This suggests that algorithm coefficients (tie points) are not representative of pond-covered ice.

General Comment

This is a long-overdue study. SIC during summer has long been issue with PM products. While there have been some comparisons in the past, none as comprehensive as this. The analysis well done and very thorough. This will definitely make an important contribution to understanding of sea ice concentration products. My main comment is that it so comprehensive that it is rather difficult to wade through. Though well-written, there is just many of details and it's difficult to not get lost in the details while trying to read through.

A couple potential things to consider that may make the paper more digestable:

1. In Section 4.1, there is a lot of dense information here and the discussion of the different groups, back-and-forth, in each subsection for each melt regime, is hard to follow. There is a summary section, but that is nearly as long as all of the individual subsections and seems to mostly restate what was said before. One thought is that instead breaking up the subsections by melt regime (pre-melt, melt advance, etc.), break up the section by algorithm groups. Then for each group go through the melt regimes.

Then in the Summary section (4.1.5), bring the different groups together to intercompare. This shows the results in two different ways rather than simply restating the results in the same way.

2. Another thing that is that ASI is part of group III. I understand the reason why in terms of the algorithm formulation. However, repeatedly in the text it is noted that ASI results are more similar to groups I and II. So, while going through all the groups, I was repeatedly having to refer/think back to groups I and II. And it made for added complexity. Perhaps it would be better to group by their characteristics in the comparison with MODIS.

3. While I commend the thoroughness of the analysis using 10 different algorithms, there are a lot of commonalities between algorithms and I wonder if maybe focusing on some of the algorithms might be more beneficial. This is in practice done to some degree by splitting them into groups and in some places showing group averages (e.g., Figures 4 and 5). But then the analysis still delves into individual algorithm products, which can be confusing. For example, Figure 6 shows only one product from each group, which are meant to be representative, yes? But then why not just use those 4 algorithms instead of 10? The SICCI algorithm for Group I is basically the same – it's just the input TB source and the spatial resolution, right? Similarly, for Group II, they are all largely some implementation of the CBT (the NOAA CDR includes NT1, but its contribution is small), so maybe just use one. For Group III, as noted above ASI doesn't really fit with NT1 and seems to mostly need to be separated out in the discussion.

I can see the value of having all algorithms, but it gets confusing keeping track of which algorithm or which group is specifically being discussed. Perhaps focus on say the four (or five with ASI) representative algorithms in the main manuscript and then compare the other algorithms within each group in the Supplement.

4. The Supplement right now consists of just extra figures. That's okay – it certainly saves some space in the paper. But I think the Supplement could serve as a useful "further discussion/analysis" document, with discussion, where some things could be further discussed, such as in #3 above, and below.

5. While I like the ship-based observation comparison (4.3) – it's a good alternative validation approach. But to me it seems like a diversion from the main thrust of the paper. So this could be something to consider moving to the Supplement. The main rationale in my mind for the section is to address the question of "but how accurate are the MODIS SIC and ISF fields?" That's an important question, but again one that might

be better treated in a supplement.

I can see where moving material to the Supplement and doing some re-structuring is not trivial, but I think it could be done without too much effort. And while I would suggest doing this for better readability and understanding, the current format is correct scientifically and the analysis is thorough. So, I would consider these minor revisions.

Specific Comments (by line number):

88: What is the "1)"? I don't see a "2)" or beyond – was something left out or is the "1)" extraneous?

139: "these values are not accessible to the user" is a repeat of what is stated in Line 136.

187-188: Why is the ratio of 1 used as a criteria for exclusion? Why does the high SD in the 500 m values indicate MPF that shouldn't be used?

198: What does "converted into Cartesian coordinates" mean? I think this effectively a drop-in-the bucket re-gridding – is that correct?

202-205: I wonder how the 8-day average affects the analysis? Melt onset can occur quite rapidly – within a day. So, what happens when the melt onset (or transition to other melt regimes) happens within the middle of an 8-day period? I understand the rationale – the MODIS product is an 8-day average – so it makes sense to do it this way. But some discussion of the ramifications (or lack thereof) may be warranted.

599: "surprising"

635-637: the sentence is a little confusing as written, instead of "the number 12 results from" maybe "the 12 fixed TB values are based on"

739: "that" instead of "which"

---

## Author Comment (AC1) · 14 May 2020

Response to the review of anonymous reviewer #1: tc-2020-35-RC1 of the manuscript tc-2020-35:

Satellite Passive Microwave Sea-Ice Concentration Data Set Intercomparison for Arctic Summer Conditions by Kern et al.

The manuscript is interesting and useful for the sea ice research. It is very useful to make this kind of comparisons. Comparisons to MODIS SIC give an idea of the weaknesses of the existing algorithms and also gives tools to correct or improve the

current algorithms, especially in the summer conditions with the highest uncertainties. The suggested bias correction looks like a promising approach to get more accurate SIC or ISF estimates. Also the potential explanations for the algorithm behaviour in summer conditions are interesting and useful information and give good information for further research and algorithm development.

» We thank the reviewer for the positive impression given about the manuscript and are grateful to the comments helping to finalize the manuscript for publication.

I only have a few minor comments: 1) The algorithms have been divided into four groups. I think the division has been in more detail reasoned in the previous TC paper of the authors. However, I would miss a short description/reasoning also here pointing out what are the actual differences between the algorithm groups and what are the similarities (and differences) within groups. I think this can be seen as a clustering analysis i.e. looking for an optimal set of clusters simultaneously minimizing within-cluster distance and maximizing between cluster difference for a set of (selected) features.

» We thank the reviewer for this comment. We added a short reasoning of the assignment of algorithms to groups in Section 2.1.

2) There are formulas of the parameters discussed given. Even though they are mostly quite simple, formulas would make the presentation even more clear, e.g. SIA = nintergral SIC dA or something similar, now they have (only) been described in words.

» We thank the reviewers and provided formulas were appropriate.

3) Some abbreviations are not explained, at least PR and GR should be explained as they appear for the first time (even though they are clear for the most readers). Still check all of these. Possibly even (general) formulas for PR and GR could be given? Also open OSI SAF and SICCI.

» Thank you. We checked the entire manuscript with respect to abbreviations not yet explained and added the respective explanation when introducing the abbreviations.

We also provided formulas for PR and GR.

4) Consider of replacing "<" and ">" in the text by "less than" and "greater/more than"

» We changed the manuscript according to this suggestion.

5) P2 L59: "...early in the melt season or on land-fast sea ice..." When this may occur over land-fast ice? Also during early melt season?

» Thank you. We replaced "or on land-fast sea ice" by "and on particularly level such as land-fast sea ice".

6) P3 L125: "... with the size of the field-of-view of..." Could this be e.g. diameter (I think the unit of the size should be km2)?

» We replaced "size" by "diameter"; thank you.

7) P7 L1149-150: "last accessed October 12, 2016". This is almost 4 years ago, update this.

» This data set hasn't changed since then. Still we updated the access date to a more recent date.

8) P5 L187-189: "...excluded all the samples..." "...larger than 1". The ratio mean(MPF)/std(MPF) is the signal to noise ratio, if You exlude the values with high SNR then You only include the uncertain data? Or did You mean the coefficient of variation std(MPF)/mean(MPF) instead?

» We thank the reviewer for the careful review. In fact, we exclude all samples where the ratio mean(MPF)/std(MPF) is SMALLER than 1. We changed the text accordingly such that it reads "smaller" instead of "larger" in Line 188.

9) Conversion to Cartesian coordinates is mentioned. But is there a certain projection You are using. What are the units in the coordinate system (e.g. meters in which projection)? This is not very clear to me based on the manuscript.
» Thank you. We replaced "For this step the coordinates of both data sets, i.e. the PMW products and the MODIS products, are converted into Cartesian coordinates ..." by "For this step, we converted the latitude and longitude coordinates of both data sets, i.e. the PMW products and the MODIS products, into metric coordinates using the WGS84 ellipsoid ..."

10) You use the term Day of the Year, it is also often referred as the Julian day. I do not know which one is better in scientific articles, I have seen both practices.

» Well, in fact we kept day of the year (DOY) then. We believe that day of the year is actually less abstract than "Julian Day".

11) P13 L534: "values held in Table 3...". Probably "values given in Tables 3 and 4" or "values held by tables 3 and 4" would sound better?

» We opted for "values given in Tables 3 and 4" in the new manuscript.

12) P15 L613: "accord" -> "accordance"?

» Changed accordingly.

13) P17 L697: "The influence different surface properties exert on the" -> "The influence exerted bydifferent surface properties..."?

» Changed as suggested.

14) Possibly You could also mention the bias correction approach of section 4.4. also in the final conclusion section (5.5.).

» We thank the reviewer for this suggestion. We discussed this issue and now mention the possibility of a bias correction towards a PMW netISF as an added value of our results in the conclusions. At the same time, we mention that a bias-correction towards true SIC is not recommended because such a result would be ambiguous, because we cannot distinguish between water on top of ice and between ice floes and because a correction towards the PMW netISF is physically more meaningful. We also added a

comment that further research on melt-pond fraction data sets is required.

---

## Author Comment (AC2) · 14 May 2020

Response to the review of anonymous reviewer #2: tc-2020-35-RC2 of the manuscript tc-2020-35:

Satellite Passive Microwave Sea-Ice Concentration Data Set Intercomparison for Arctic Summer Conditions by Kern et al.

This paper compares ten passive microwave (PM) sea ice concentration algorithms during summer melt conditions in the Arctic. The comparisons are done relative to a MODIS-derived surface classification product that distinguishes open water, melt

ponds, and unponded sea ice. Ship observations are also used in the comparison. The ten algorithms are split into four characteristic groups based on the formulation of the algorithms. Comparisons are made for sea ice concentration (SIC) and net ice-surface fraction (ISF = SIC minus melt pond fraction). The results show varied performance of the algorithms during different stages of the melt period. Summer SIC is overestimated by some algorithms, but underestimated by others. However, ISF is systematically overestimated by all algorithms. This suggests that algorithm coefficients (tie points) are not representative of pond-covered ice.

General Comment

This is a long-overdue study. SIC during summer has long been issue with PM products. While there have been some comparisons in the past, none as comprehensive as this. The analysis well done and very thorough. This will definitely make an important contribution to understanding of sea ice concentration products. My main comment is that it so comprehensive that it is rather difficult to wade through. Though well-written, there is just many of details and it's difficult to not get lost in the details while trying to read through.

» We thank the reviewer for the positive perception of our manuscript and the very help comments to, as the reviewer stated, make the manuscript more "digestable".

A couple potential things to consider that may make the paper more digestable: 1. In Section 4.1, there is a lot of dense information here and the discussion of the different groups, back-and-forth, in each subsection for each melt regime, is hard to follow. There is a summary section, but that is nearly as long as all of the individual subsections and seems to mostly restate what was said before. One thought is that instead breaking up the subsections by melt regime (pre-melt, melt advance, etc.), break up the section by algorithm groups. Then for each group go through the melt regimes. Then in the Summary section (4.1.5), bring the different groups together to intercompare. This shows the results in two different ways rather than simply restating the results in the

same way.

» We thank the reviewer for this very constructive suggestion. We were thinking about this suggestion. We definitely see the point of the reviewer with respect to the mentioned back-and-forth between groups for each melt regime. This could call for indeed for re-ordering subsections such that these are referring to groups rather than seasons.

However, at the same time, by keeping the subsections as they are we have the possibility to refer to observations made for two or more groups in one go. Since we quite often have similar observations for group I and III on the one hand and group II and IV on the other hand, we benefit from this advantage quite often. Describing results for each group separately would – to our opinion – increase the amount of repetitions.

We therefore propose to keep the sections as they are but reduce the complexity of the information given and keep an eye on reducing the back-and-forth between the groups. We also reduced the amount of text where we simply describe the figures.

2. Another thing that is that ASI is part of group III. I understand the reason why in terms of the algorithm formulation. However, repeatedly in the text it is noted that ASI results are more similar to groups I and II. So, while going through all the groups, I was repeatedly having to refer/think back to groups I and II. And it made for added complexity. Perhaps it would be better to group by their characteristics in the comparison with MODIS.

» We thank the reviewer. This is for sure a valid point as well and was recognized by the authors during the writing. However, we indeed wanted to benefit from the assignment of algorithms into groups from the companion earlier paper in The Cryosphere. Since we also refer to the more detailed descriptions of the algorithms as well as groups in that earlier paper, we believe that it would cause considerable confusion between the two papers (and a third one to come).

Responding to reviewer #1 we added a short paragraph to section 2.1 in which we

repeat our reasoning of assigning products to groups. In this context, we now make a note that, as will be demonstrated throughout the paper, the ASI product is assigned to group III technically, as laid out in Kern et al., 2019, but that actually during summer conditions, this product appears to be belonging more closely to group I. We then do not refer to this issue anymore before it comes to the discussion of Figure 13 (in the original submission). In that figure it appears obvious that ASI behaves different to NT1 and in that context we are then going to refer to the "full" (or original) versions of Figures 4, 7 and 10 which are now in the supplementary material.

3. While I commend the thoroughness of the analysis using 10 different algorithms, there are a lot of commonalities between algorithms and I wonder if maybe focusing on some of the algorithms might be more beneficial. This is in practice done to some degree by splitting them into groups and in some places showing group averages (e.g., Figures 4 and 5). But then the analysis still delves into individual algorithm products, which can be confusing. For example, Figure 6 shows only one product from each group, which are meant to be representative, yes? But then why not just use those 4 algorithms instead of 10? The SICCI algorithm for Group I is basically the same – it's just the input TB source and the spatial resolution, right? Similarly, for Group II, they are all largely some implementation of the CBT (the NOAA CDR includes NT1, but its contribution is small), so maybe just use one. For Group III, as noted above ASI doesn't really fit with NT1 and seems to mostly need to be separated out in the discussion.

I can see the value of having all algorithms, but it gets confusing keeping track of which algorithm or which group is specifically being discussed. Perhaps focus on say the four (or five with ASI) representative algorithms in the main manuscript and then compare the other algorithms within each group in the Supplement.

» We thank the reviewer. Actually, our motivation for showing only 4 of the 10 products was driven by the observation (from this paper and from Kern et al., 2019) that it appears to be sufficient to show and discuss the results of one representative product / algorithm throughout the majority of the paper. This is a compromise, however. While

we agree that "Just the TB input source" is changed, it is exactly one of the main points, which needs to be stressed and discussed – both for the existing products as well as for future product development. This applies primarily to group I but also to group III (where the main SIC information comes from a polarization difference). What we find worth noting is also the "mediating" influence of NT onto the NOAA-CDR product.

We try to avoid putting information into supplementary material; even though this is only one additional click away it is often this additional click, which is not happening and with that often an important part of the paper is not digested by the reader. Therefore, we are reluctant to put important parts of the interpretation or discussion into the supplementary material.

What we now plan doing as a compromise is the following: We show Figures 4, 7 and 10 as they appear in the original submission in the supplement (and refer to these from the main text) and reduce these figures in the main text to a modified version, which in fact only show the 4 "representative" algorithms.

4. The Supplement right now consists of just extra figures. That's okay – it certainly saves some space in the paper. But I think the Supplement could serve as a useful "further discussion/analysis" document, with discussion, where some things could be further discussed, such as in #3 above, and below.

» We thank the reviewer for this view but we would like to refer to our reply to your comment #3 with respect to our view of the content of supplementary material. We will put more information there, yes, and see this as a compromise required to reduce the content of the paper.

5. While I like the ship-based observation comparison (4.3) – it's a good alternative validation approach. But to me it seems like a diversion from the main thrust of the paper. So this could be something to consider moving to the Supplement. The main rationale in my mind for the section is to address the question of "but how accurate are the MODIS SIC and ISF fields?" That's an important question, but again one that might

be better treated in a supplement.

» We thank the reviewer for this constructive suggestion and actually agree with it under the auspices that this is "just" an additional alternative way to look. We moved section 4.3 and the associated other parts (section 2.3 and fig. 3 as well as the relevant references) into the supplementary material.

I can see where moving material to the Supplement and doing some re-structuring is not trivial, but I think it could be done without too much effort. And while I would suggest doing this for better readability and understanding, the current format is correct scientifically and the analysis is thorough. So, I would consider these minor revisions.

» We thank the reviewer for bringing up the above-mentioned suggestions and hope that we have made the paper easier to digest with the new version.

Specific Comments (by line number): 88: What is the "1)"? I don't see a "2)" or beyond – was something left out or is the "1)" extraneous?

» We removed "1)".

139: "these values are not accessible to the user" is a repeat of what is stated in Line 136.

» We thank the reviewer. We deleted the sentence starting in Line 136: "However, the above-100% . . . for validation"

187-188: Why is the ratio of 1 used as a criteria for exclusion? Why does the high SD in the 500 m values indicate MPF that shouldn't be used?

» We thank the reviewer. This issue was also marked by reviewer #1. First of all, in fact, we excluded all samples where the ratio mean(MPF)/std(MPF) is SMALLER than 1. We changed the text accordingly such that it reads "smaller" instead of "larger" in Line 188. Secondly, by including only data where this ratio (which is actually the signal-to-noise ratio) is larger than 1 we assure that cases with low MPF but high variability are
excluded as these cases most often belong to cases with artificially high MPF caused, e.g. by cloud shadows.

198: What does "converted into Cartesian coordinates" mean? I think this effectively a drop-in-the bucket re-gridding – is that correct?

» We thank the reviewer; here we were not referring to the gridding of the 500 m resolution pixels into the final gridded MPF product but we were referring to the step required for an adequate co-location of the data. Reviewer #1 commented on this issue as well. We replaced "For this step the coordinates of both data sets, i.e. the PMW products and the MODIS products, are converted into Cartesian coordinates . . ." by "For this step, we converted the latitude and longitude coordinates of both data sets, i.e. the PMW products and the MODIS products, into metric coordinates using the WGS84 ellipsoid . . ."

202-205: I wonder how the 8-day average affects the analysis? Melt onset can occur quite rapidly – within a day. So, what happens when the melt onset (or transition to other melt regimes) happens within the middle of an 8-day period? I understand the rationale – the MODIS product is an 8-day average – so it makes sense to do it this way. But some discussion of the ramifications (or lack thereof) may be warranted.

» We thank the reviewer and see the necessity to provide some explanation. Actually, the 8-day product is not an average but a composite of clear-sky images. If a region experience clear-sky conditions on three days of the 8-days period it is the most recent clear-sky case which counts.

In addition, as we wrote, the transitions between the four melt regimes are not based on a particular thresholds but are defined via the pan-Arctic average development of melt-pond cover (as displayed in Figures 2, 7 and 10 of the original submission). Therefore, we are sure that variations in the actual melt onset within the 8-days periods has little effect on our results and is possibly reflected as noise (e.g. the standard deviations in the MPF curves in Fig. 7 and 10 of the original submission).

[Figure]

More than melt-onset on different days within an 8-days period do regional differences in melt onset influence our results. We believe, however, that an investigation of local differences in melt onset and their impact on our results are beyond the scope of this paper.

We would like to refer to the paper by Kern et al., The Cryosphere, 2016, in which we used three months (June through August) of daily MODIS melt-pond cover fraction data set and which provided similar results in terms of the various melt stages.

We decided to add more information about the reliability / limitations of this 8-days MODIS data set in the context of its description in Section 2.2.

599: "surprising"

» Corrected as suggested.

635-637: the sentence is a little confusing as written, instead of "the number 12 results from" maybe "the 12 fixed TB values are based on"

» Changed as suggested.

739: "that" instead of "which"

» Changed as suggested.